# Towards Reliable Misinformation Mitigation: Generalization, Uncertainty, and GPT-4

**Kellin Pelrine[1], Anne Imouza[1], Camille Thibault[2], Meilina Reksoprodjo[3],**
**Caleb A. Gupta[4], Joel N. Christoph[5], Jean-François Godbout[2], Reihaneh Rabbany[1]**
[1]McGill University; Mila    [2]Université de Montréal    [3]Eindhoven University of Technology
[4]University of Pennsylvania    [5]European University Institute

## Abstract

Misinformation poses a critical societal challenge, and current approaches have yet to produce an effective solution. We propose focusing on generalization, uncertainty, and how to leverage recent large language models, in order to create more practical tools to evaluate information veracity in contexts where perfect classification is impossible. We first demonstrate that GPT-4 can outperform prior methods in multiple settings and languages. Next, we explore generalization, revealing that GPT-4 and RoBERTa-large exhibit differences in failure modes. Third, we propose techniques to handle uncertainty that can detect impossible examples and strongly improve outcomes. We also discuss results on other language models, temperature, prompting, versioning, explainability, and web retrieval, each one providing practical insights and directions for future research. Finally, we publish the LIAR-New dataset with novel paired English and French misinformation data and Possibility labels that indicate if there is sufficient context for veracity evaluation. Overall, this research lays the groundwork for future tools that can drive real-world progress to combat misinformation.

## 1 Introduction

Misinformation represents a significant societal challenge, with detrimental effects observed in various domains spanning elections (Meel and Vishwakarma, 2020), public health (Loomba et al., 2021), the economy (Marcelo, 2023), and more. Recent developments in generative models have enabled the creation of increasingly sophisticated and convincing AI-generated misinformation, including deepfakes and bots using large language models, which further exacerbate the potential harm of "fake news" (Zhou et al., 2023). As a result, developing systems to limit the spread and impact of misinformation is of critical importance. Although there is a great deal of research in this area (Shu

et al., 2017; Sharma et al., 2019; Shu et al., 2020; Kumar et al., 2021; Shahid et al., 2022), practical solutions remain elusive. Could the recent progress in generative language models provide a path to reliable veracity evaluation?

In this work, we investigate the predictive capabilities of GPT-4 compared to previous approaches. We propose going beyond the focus on direct classification performance that is common in this domain and also prioritizing understanding of generalization and uncertainty. We believe this is critical since, despite extensive prior work on misinformation detection, existing solutions often fail to generalize and work in real-world settings, and there is no path to a perfect classifier that will solve this challenge with oracle-like answers alone. Therefore, a better understanding of generalization will help shrink the gap between research and deployment. Similarly, a better understanding of uncertainty will enable systems to fail gracefully, informing users and practitioners when predictions are certain enough to be relied upon, and providing useful information without requiring impossibly perfect accuracy. Overall, this paper aims to lay groundwork that will lead to more practical systems and, hopefully, improvements in real-world misinformation mitigation.

We examine three datasets: the widely-used LIAR (Wang, 2017), CT-FAN-22 (Köhler et al., 2022) that contains both English and German corpora, and a new dataset LIAR-New. The latter was constructed to provide data beyond GPT-4's main knowledge cutoff, to include examples in English and French, and to understand whether examples have sufficient context for evaluating their veracity. We show superior classification results with GPT-4 compared with the literature, both directly and in transfer and multilingual settings. We analyze the errors made by GPT-4 and find they are quite different from a standard RoBERTa approach. We also conduct extensive experiments on uncer-

tainty and context, examining ways to productively quantify uncertainty and evaluate how results are affected by insufficient input information. One of the approaches here (GPT-4 Uncertainty-Enabled) detects many Impossible examples and significantly increases performance. In addition, we show how extra label information can improve existing approaches, and that hard classification performance does not necessarily predict soft (probabilistic) performance. Finally, we provide preliminary analysis of web retrieval, showing that it provides a path to greatly improving performance, and explainability, showing it yields insights that motivate future work.

In summary, our key contributions are the following:

- We show that GPT-4 can work for both hard and soft classification tasks for misinformation, including multilingual ones, and give better performance than existing approaches.
- We conduct extensive analyses on the generalization of GPT-4 in relation to prior work, finding that the errors made are substantially different and that GPT-4 has superior generalization.
- We propose a method to account for uncertainty with GPT-4 that excludes examples it cannot classify well and improves results by 8 percentage points on LIAR-New and 14 on LIAR.
- We conduct a number of experiments aimed at important practical factors or showing paths to build on this work, including other language models, temperature, prompting, versioning, explainability, and web retrieval.
- We release a new dataset, LIAR-New, which has novel English plus French data and Possibility labels. We demonstrate how this data can be used to better understand generalization and contextual uncertainty.

**Reproducibility**: We open-source our code and other contributions on Github.[1]

## 2 Data

We experiment on several misinformation datasets. First, we analyze the LIAR dataset (Wang, 2017), which is one of the most widely-used benchmarks for fake news detection. It contains nearly 13K examples with veracity labeled on a 6-point scale. We mainly follow the common approach in the literature of binarizing these labels, but also report

6-way results. Second, we use the CT-FAN-22 dataset (Köhler et al., 2022) for additional tests, including transfer and multilingual settings. It contains an English and a German corpus with 4-way labels, 3 of which represent gradations of veracity and an "Other" category for uncertain or unverified examples. Finally, we provide a new dataset, LIAR-New. This dataset goes beyond the GPT-4 main knowledge cutoff of September 2021. It also provides novel contributions to the misinformation literature: first, every input is provided in both the original English language version and in French through human translation. Second, it includes Possibility labels that assess if an example is missing the necessary context and information to determine whether it is misinformation or not. We discuss the first two datasets in detail in Appendix B, and our new dataset below.

### 2.1 LIAR-New

To minimize the possibility of test label leakage with GPT-4—i.e., to ensure that it is not relying on directly seeing the labels somewhere in its training data—we scraped PolitiFact for recent fact-checks that are beyond the main GPT-4 knowledge cutoff. There is no 100% fixed endpoint, but "the vast majority of its data cuts off" in September 2021 (OpenAI, 2023). We scraped all fact-checks between IDs 21300 and 23300, which represents roughly the end of September 2021 to mid-November 2022. In our experiments we remove the 27 examples from September 2021, leaving only data from October 1st, 2021 and beyond.

The form and source of the statements are the same as LIAR, as well as the labels, with the exception that PolitiFact changed the name of the class "Barely-true" to "Mostly-false". We note, however, that the class balance is very different: this dataset contains far more False examples than any other classes, and does not contain many True ones. This could correspond, for example, to a real-world scenario where users of a system are mostly interested in fact-checking statements that seem suspicious. Here, we do not train on this dataset, rather we use it solely for evaluation purposes.

In addition to experiments on the English statements provided by PolitiFact, we also translated all examples into French. This was done by two authors who are native French speakers and also fluent in English. The translation was done without the aid of machine translation, in order to avoid

[1] https://github.com/ComplexData-MILA/MitigateMisinfo

| | Pants-fire | False | Mostly-false | Half-true | Mostly-true | True | Total |
|---|---|---|---|---|---|---|---|
| **Test** | 359 | 1067 | 237 | 147 | 99 | 48 | 1957 |

Table 1: LIAR-New dataset statistics.

potential biases that might make the task easier for systems that incorporate such translation. Although this is not the first misinformation dataset in French, there are not many, and we struggled to find one that is easily accessible and has clear provenance and quality. Thus, this French data may be practical both in and of itself, and for comparative analysis with the English pairing. We make the full data available on Zenodo.[2]

## 2.2 Possible vs. Impossible Examples

Not all inputs can be evaluated for veracity because certain claims cannot be verified without the proper context. To better understand this problem in relation to our data, we define the following labels for a given statement:

**Definition 1.** *Possible: the statement's claim is clear without any additional context, or any missing context does not make it difficult to evaluate the claim.*

*Hard: the claim is missing important context that makes it hard to evaluate, but it might still be possible.*

*Impossible: there is missing context that cannot be resolved ("The senator said the earth is flat"— not possible without knowing which senator). Or, the statement contains no claim for which veracity could be evaluated regardless how much context is available.*

It is important to note that, in an extreme sense, no statement can be verified without some level of additional context (for example, a speaker could reference an imaginary world in which the sun *does* orbit the earth). Therefore, these definitions are not meant to be an ironclad, encyclopedic formulations; they is instead intended to capture a reasonable standard whereby a model with training knowledge—but without additional context for a given statement—can reasonably evaluate that statement's veracity. They aim to strike a balance between practicality and accuracy, allowing for the effective categorization of statements containing misinformation while acknowledging the inherent complexity and context-dependency of language.

Three annotators each labeled the entire dataset

[2]https://zenodo.org/records/10033607

according to this definition. After a process of disagreement resolution, this yielded 927 Possible, 581 Hard, and 449 Impossible examples. The numerous Impossible examples show that such analysis is critical—training and evaluating on these examples unknowingly could lead to results that will not generalize well to real-world systems. In the Experiments section we examine the different cases. We present more detailed definitions and information on the labeling process in Appendix C.

## 3 Methodology

### 3.1 GPT-4

**Main Prompting** To test the effectiveness of GPT-4 for both hard and soft classification, we crafted a concise prompt asking GPT-4 to rate the truthfulness of a statement on a scale of 0 to 100, without providing any explanations. We denote the approaches using this prompt as "GPT-4 Score".

> "role": "user", "content": 'Rate the truthfulness of the following statement: "STATEMENT" Provide a score from 0 to 100, where 0 represents definitively false and 100 represents definitively true. Do not provide any explanations, only respond with the numerical score.'

We also test a binary classification prompt, which we tried to make as similar as possible to the score prompt above. We denote this approach as "GPT-4 Binary".

> "role": "user", "content": 'Rate the truthfulness of the following statement: "STATEMENT" Provide a number 0 or 1, where 0 represents false and 1 represents true. Do not provide any explanations, only respond with the number.'

Finally, we examine a prompt that explicitly allows for GPT-4 to refuse to pass a judgment. It was constructed from the above binary prompt based on the observation that GPT-4 sometimes responded with a score of "0.5", despite the prompt instructing it to respond 0 or 1 only (see also discussion below). We denote this approach "GPT-4 Binary Uncertainty-Enabled".

> "role": "user", "content": 'Rate the truthfulness of the following statement: "STATEMENT" Provide a number 0 or 1, where 0 represents false and 1 represents true. If you are uncertain or there is not enough context in the statement to be sure what it refers to, instead answer 0.5. Do not make assumptions. Do not provide any explanations, only respond with the number.'

**Other Prompting** We experimented with several other types of prompts. Explainability prompts are discussed in Section 4.8 and Appendix Q. Web retrieval prompts are discussed in Section 4.9 and Appendix R. Several attempts at in-context learning with automatically chosen demonstration are detailed in Appendix D.

**Other GPT-4 Implementation Details** In most of our experiments, we opted for a moderate temperature value of 0.5 in the API to balance reproducibility and noise against nuance and model effectiveness. We later found that 0.0 temperature seems to yield better results, as discussed in Appendix J,. Due to time and cost limits, however, we did not rerun the other experiments. To ensure that the full output was captured in cases where it did not return a number, we set the "max tokens" value to 1000. Cost estimates are available in Appendix I.

To convert the GPT-4 output to predictions we could evaluate, we first separated integer and non-integer predictions. We found the latter cases generally occurred when GPT-4 refused or stated it was incapable of evaluating the veracity of the input. With GPT-4 Score, in the LIAR dataset, this occurred 40 times in the validation set and 69 times in the test set, out of 1284 and 1267 examples respectively. In CT-FAN-22 it occurred 2 times on English data and 0 on German. In LIAR-New, it occurred only once.

One of the authors manually examined the LIAR test set cases. According to Definition 2, all were Impossible without additional information. These included statements like "On [...]" that seem to be a recurring PolitiFact headline and have no veracity on their own to evaluate: e.g., "On banning earmarks" or "On redistricting". Other examples GPT-4 refused to answer would require in-depth speaker and context information, such as "I said no to these big bank bailouts" (one would need to know the speaker and which big bank bailouts they are talking about). Therefore, on these 69 examples, GPT-4 has perfect precision in determining that these statements cannot be evaluated in isolation.

In the Experiments section, when not otherwise stated, we nonetheless make a prediction for these examples in the GPT-4 results by selecting a label uniformly at random. This ensures our results are fully comparable to other studies in the literature. Separately, we further investigate these examples in relation to other models.

To convert GPT-4 Score's 0-100 predictions to hard classification predictions, we examine two approaches. First, *GPT-4 Score Zero-Shot* simply divides the range evenly (e.g., for the binary case, splitting at 50). This is arbitrary but intuitive. Second, on binary LIAR, we tested *GPT-4 Score Optimized* which uses the threshold that gives optimal weighted F1 on the validation set. There, we found that the optimal threshold is 71—GPT-4 appears to be biased towards predicting "True" compared to the labels in the LIAR dataset. Note that neither of these (nor GPT-4 Binary) use any of the training data, and even GPT-4 Score Optimized does not maintain any memory of the validation set inputs beyond this threshold.

For testing on non-English data, we keep everything unchanged, including leaving the prompt in English.

## 3.2 RoBERTa-large

Prior work, such as Pelrine et al. (2021) and Truică and Apostol (2023), has shown that older, mostly non-causal (bidirectional) transformer-based language models can give strong performance for misinformation classification. We used the code made public by Pelrine et al. (2021) to implement RoBERTa-large (Liu et al., 2019), which had the best performance in their study. We trained the model for 10 epochs with the given hyperparameter configuration and other implementation details, using an RTX8000 GPU. For conciseness in tables, we refer to this model as RoBERTa-L Binary or RoBERTa-L 6-way, depending on whether it was trained with binary or 6-way labels. We focus our error analysis and a number of other experiments on this model compared to GPT-4, because by using the existing code we can ensure its implementation is unbiased, and it has already shown consistently strong performance on a number of datasets in the literature.

## 3.3 Other Models

**Other Small Language Models** We also fine-tuned a selection of SLMs besides RoBERTa. Implementations are detailed in Appendix E and results are reported in several experiments.

**Fuzzy Clustering** We performed limited tests with Fuzzy Clustering but the performance was not promising. We discuss this method in Appendix F.

**GPT-3.5** This model gave solid performance on LIAR, but it appeared brittle and did not perform

well on other datasets. We discuss the results in Appendix G.

**PaLM 2 Bison**  We conducted a limited evaluation of PaLM 2 (Anil et al., 2023), and found that the second largest version gives passable, though not impressive, performance on LIAR. The largest version was not accessible to us as of this writing. We discuss this model and the results in Appendix O.

### 3.4 Evaluation Metrics

We note our evaluation metrics in Appendix H.

## 4 Experiments

### 4.1 LIAR Binary Classification

In Table 2 we present binary classification results on the LIAR dataset (6-way classification results are presented in Appendix N). We see here that GPT-4 Score Optimized performs best, suggesting it can be a powerful new tool for misinformation detection. GPT-4 Score Zero-Shot is significantly worse but still better than several approaches that rely on the training data. GPT-4 Binary is in between the two, equal or better than all prior approaches.

We also find that some of our other language model approaches give performance exceeding previous state-of-the-art methods in the literature, paralleling older findings of Pelrine et al. (2021). Finally, we note that training RoBERTa-L on 6-way labels gives better performance than training on binary labels. This indicates the 6-way labels provide additional helpful information even when the goal is binary prediction. There is a parallel here with the work of Matsoukas et al. (2020) in other domains.

### 4.2 CT-FAN-22 Classification

We next present in Table 3 our results on CT-FAN-22. We compare zero-shot GPT-4, RoBERTa-large trained on 6-way LIAR classification (converted to 3-way output here by dividing the 6 labels evenly in 3), and the previous SOTA. We see that GPT-4 has a clear advantage in this setting, both in English and in German. RoBERTa performs poorly. This provides additional evidence for the strength of GPT-4 in general compared to models trained on in-distribution data. It also demonstrates that this model provides a large advantage in transferability to new data, and shows it can also work in multilingual settings better than previous approaches.

| Method | Accuracy | F1 |
|---|---|---|
| SOTA (2022; 2021) | 62 | - |
| GPT-4 Score Optimized | **68.2** | **68.1** |
| GPT-4 Score Zero-Shot | 64.9 | 60.9 |
| GPT-4 Binary | 66.5 | 66.5 |
| RoBERTa-L Binary | 63.5 | 62.1 |
| RoBERTa-L 6-way | 64.7 | 64.1 |
| BERT | 65.0 | 64.5 |
| ConvBERT | 66.7 | 65.8 |
| DeBERTA | 63.0 | 63.8 |
| DeBERTA-V3 | 65.0 | 64.4 |
| LUKE | 65.3 | 64.4 |
| RoBERTa | 64.7 | 64.2 |
| SqueezeBERT | 63.1 | 62.2 |
| XLMRoBERTA | 61.1 | 61.0 |
| Fuzzy (Word2Vec) | 60.2 | 60.4 |
| Fuzzy (BERT) | 59.6 | 60.1 |
| Fuzzy (GloVe) | 60.1 | 59.7 |

Table 2: Binary Classification Results (percentages). GPT-4 shows superior performance. For RoBERTa-L, even though the test is binary classification, training on 6-way labels provides an advantage.

| | Method | Accuracy | F1 |
|---|---|---|---|
| **English** | SOTA (2022) | 54.7 | 33.9 |
| | GPT-4 Score Zero-Shot | **67.8** | **42.8** |
| | RoBERTa-L 6-way | 47.5 | 26.8 |
| **German** | SOTA (2022) | 42.7 | 29.0 |
| | GPT-4 Score Zero-Shot | **57.6** | **38.7** |

Table 3: 4-way classification in a transfer setting. On English data, GPT-4 beats the SOTA approach trained on in-distribution data, while RoBERTa trained on LIAR degrades. On German data, where there is no in-distribution training data available, GPT-4 again beats SOTA—even without changing the prompt, which is written in English and makes no mention of German.

In Table 4, we remove the "Other" label category, which is not well-defined and for which we automatically marked all GPT-4 and RoBERTa predictions as wrong. Performance improves by quite a large margin. These results set the bar for future work on this dataset with the "Other" label excluded, which we recommend, as discussed in Section B.2.

### 4.3 LIAR-New Classification

We next examine new data beyond the main GPT-4 knowledge cutoff. We see in Table 5 that it can nonetheless make effective predictions. This suggests the model is taking advantage of its overall knowledge to make predictions on examples related to older events and facts (and perhaps make educated guesses on newer ones), rather than just

|         | Method | Accuracy | F1 |
|---------|--------|----------|-----|
| **English** | GPT-4 Score Zero-Shot | 71.4 | 58.6 |
|         | RoBERTa-L 6-way | 50.0 | 36.5 |
| **German** | GPT-4 Score Zero-Shot | 63.6 | 54.3 |

Table 4: 3-way classification in a transfer setting, excluding the label "Other" which is not well-defined. GPT-4 again beats RoBERTa.

seeing the label during its training. This also suggests that GPT-4 has great potential if it could be combined with an effective web search to evaluate examples that require recent knowledge that it is missing. We also find that:

- GPT-4 Binary again outperforms GPT-4 Score Zero-Shot, suggesting the score the latter predicts is not quite aligned with the labels in this binarization.
- RoBERTa-L 6-way is again better than RoBERTa-L Binary, further confirming the benefit of training on 6-way labels.
- GPT-4's performance decreases significantly as we move from Possible to Hard to Impossible examples, while RoBERTa has the opposite trend. This suggests GPT-4 in many cases may be relying on a generalizable information to assess veracity, while RoBERTa may be taking advantage of distributional patterns that provide cues in this dataset but won't generalize (as we saw in the preceding transfer experiment).
- GPT-4 gives superior performance overall.

|         | Method | Accuracy | F1 |
|---------|--------|----------|-----|
| **Possible** | GPT-4 Binary | **91.0** | **73.7** |
|         | GPT-4 Score Zero-Shot | 82.9 | 61.5 |
|         | RoBERTa-L 6-way | 79.4 | 57.3 |
|         | RoBERTa-L Binary | 71.9 | 55.3 |
| **Hard** | GPT-4 Binary | **74.0** | **68.8** |
|         | GPT-4 Score Zero-Shot | 64.6 | 61.1 |
|         | RoBERTa-L 6-way | 63.9 | 60.9 |
|         | RoBERTa-L Binary | 60.1 | 58.2 |
| **Impossible** | GPT-4 Binary | **71.0** | **58.5** |
|         | GPT-4 Score Zero-Shot | 56.1 | 49.9 |
|         | RoBERTa-L 6-way | 83.6 | 71.5 |
|         | RoBERTa-L Binary | 76.2 | 66.1 |
| **All** | GPT-4 Binary | **81.2** | **68.8** |
|         | GPT-4 Score Zero-Shot | 71.3 | 60.5 |
|         | RoBERTa-L 6-way | 75.8 | 64.0 |
|         | RoBERTa-L Binary | 69.4 | 60.3 |

Table 5: LIAR-New binary classification. GPT-4 shows strong performance, even beyond its main knowledge cutoff.

### 4.3.1 Misinformation in French

In Table 6, we test GPT-4 on the French version of the dataset. We see that performance decreases noticeably compared to English (i.e., Table 5). Part of this is likely due to GPT-4's overall performance in different languages, but we also hypothesize that the language can affect how GPT-4 imagines the context—for example, the best guess at who "the president" refers to might be the US president in English but the French president in French. We see that the drop in performance is a bit smaller on the Possible examples compared to the others. These are examples where there should be no need to guess the context, so this weakly supports the above hypothesis, but could warrant further investigation. Moreover, the significant drop in performance in French, which is still a relatively common language, should motivate more misinformation mitigation work in low-resource languages. As seen in the previous German experiments, GPT-4 may be doing better than previous approaches in other languages, but still has a lot of room for improvement.

|         | Method | Accuracy | F1 |
|---------|--------|----------|-----|
| **Possible** | GPT-4 Binary | 89.6 | 66.1 |
|         | GPT-4 Score Zero-Shot | 77.3 | 59.9 |
| **Hard** | GPT-4 Binary | 71.1 | 60.6 |
|         | GPT-4 Score Zero-Shot | 60.3 | 58.9 |
| **Impossible** | GPT-4 Binary | 73.9 | 53.5 |
|         | GPT-4 Score Zero-Shot | 53.5 | 46.4 |
| **All** | GPT-4 Binary | 80.5 | 61.8 |
|         | GPT-4 Score Zero-Shot | 70.1 | 57.5 |

Table 6: Results on the French translation of LIAR-New. Performance in French is lower. GPT-4 is still very capable of identifying misinformation in that language, but future work to close this gap would be beneficial.

We also observed that in the case of GPT-4 Score Zero-Shot on the French data, there were 107 examples that it refused to classify, of which 62 were Impossible and 30 were Hard. This compares to single-digit numbers where it refused for GPT-4 Binary in French, and both models in English. This difference between the two versions of GPT-4 on the French data, and likewise on French vs. English data, suggests prompt engineering can have a significant impact on examples it might refuse to classify, and in turn on how a system would work if it encountered missing context or otherwise Impossible examples in the real world.

### 4.4 Uncertainty Assessment

In Table 7, we consider the uncertainty quantification of GPT-4 Score vs. RoBERTa-L on LIAR-New. We remove all examples predicted between 49 and 51 inclusive (resp. 0.49 and 0.51) by GPT-4

(resp. RoBERTa-L's final softmax), i.e., the ones each model is the most uncertain about. We see that this significantly increases the performance of GPT-4 (compared to the respective rows of Table 5) while leaving RoBERTa-L virtually unchanged.

| | Method | Accuracy | F1 |
|---|---|---|---|
| **Possible** | GPT-4 Score Zero-Shot | 90.6 | 62.7 |
| | RoBERTa-L Binary | 72.3 | 55.6 |
| **Hard** | GPT-4 Score Zero-Shot | 72.9 | 64.3 |
| | RoBERTa-L Binary | 60.7 | 58.8 |
| **Impossible** | GPT-4 Score Zero-Shot | 69.9 | 58.5 |
| | RoBERTa-L Binary | 76.3 | 66.2 |
| **All** | GPT-4 Score Zero-Shot | 81.1 | 64.3 |
| | RoBERTa-L Binary | 80.4 | 60.7 |

Table 7: Removing the examples where each model indicated the most uncertainty on LIAR-New. GPT-4 performance improves, giving an example of productive uncertainty quantification in this context.

We next examine GPT-4 Binary Uncertainty-Enabled. It responded "0.5" to 906 total examples (i.e., indicating it was not sure about their evaluation), out of which 306 were Impossible and 352 were Hard. This seems like a reasonable result—the majority of the "0.5" cases are Impossible ones that should receive that rating or Hard ones where it is reasonable for the model to be uncertain. It is also not unreasonable for the model to be uncertain about some of the Possible cases as well, especially considering the challenge of this dataset being beyond the main knowledge cutoff. It is also detecting over two thirds of the Impossible examples, although we note around 140 remain undetected, so its judgment is not perfect. Nonetheless, when we exclude these "0.5" examples and look at the classification performance on the remaining examples (Table 8), we see a marked increase in performance compared to other approaches.

| | | Method | Accuracy | F1 |
|---|---|---|---|---|
| **LIAR-New** | **Possible** | GPT-4 Binary Uncertainty-Enabled | **94.4** | 75.1 |
| | **Hard** | GPT-4 Binary Uncertainty-Enabled | 86.5 | **78.4** |
| | **Impossible** | GPT-4 Binary Uncertainty-Enabled | 91.4 | 70.4 |
| | **All** | GPT-4 Binary Uncertainty-Enabled | 92.3 | 76.8 |
| **LIAR** | | GPT-4 Binary Uncertainty-Enabled | 82.2 | 82.0 |

Table 8: Allowing GPT-4 to refuse to answer examples it is not confident on excludes many Impossible examples and greatly improves performance on LIAR-New, as well as LIAR.

We also report in the same table the results on LIAR. Here, we do not have Possibility labels to compare with. We observed that this approach excluded 914 examples, which is considerable, approximately 75% of the dataset. Based on the LIAR-New results, it is not unlikely that a large portion of these examples are Impossible. However,

the distributions are different (LIAR was sampled to have relatively balanced classes), so labeling would be needed to confirm. Regardless, performance on the remaining data is approximately 14 percentage points better than the best performance on LIAR in Table 2.

These results show a clear ability to leverage GPT-4's assessment of its confidence in classifying examples in this domain. Indeed, GPT-4 Binary Uncertainty-Enabled gives the strongest performance of any method in this paper excluding web retrieval (which could likely be combined with Uncertainty-Enabled). Granted, this is at the cost of excluding a significant number of examples. But the majority of the ones that were excluded are Impossible, and this is a domain where wrong answers can potentially be much more costly than no answer (for example, human users of a misinformation detection system might be misled by an inaccurate machine judgment, instead of making their own judgment from external sources). We believe future work should build on this capability, especially integrating it with explainability, which GPT-4 also has natural capabilities for.

### 4.5 Error Analysis

We compare the errors made by GPT-4 Score Zero-Shot and RoBERTa-L Binary on LIAR. Excluding cases GPT-4 refused to classify (which were found to be Impossible; see Section 3.1, "Other GPT-4 Implementation Details"), we find there are 241 examples that GPT-4 correctly classifies and RoBERTa-L does not, and 174 cases where the reverse is true. On LIAR-New, when we restrict the analysis to the 911 Possible examples after October 1st, 2021, there are 182 examples that GPT-4 correctly classifies and Roberta-L does not, and 82 cases the reverse. In both datasets, this represents a sizeable proportion, so it indicates a substantial difference in these models' failure modes.

We further investigated this by embedding examples with OpenAI's Ada-002 model (Greene et al., 2022). We found a statistically significant difference (p=0.0005) in embedding space distances between train and test examples, depending on whether GPT-4 was correct and RoBERTa-L wrong, or vice versa. These results suggest methods to help GPT-4 leverage the training data could lead to a nearly 10 percentage point improvement. More detailed discussion can be found in Appendix L.

In Appendix M, we present a case study of ex-

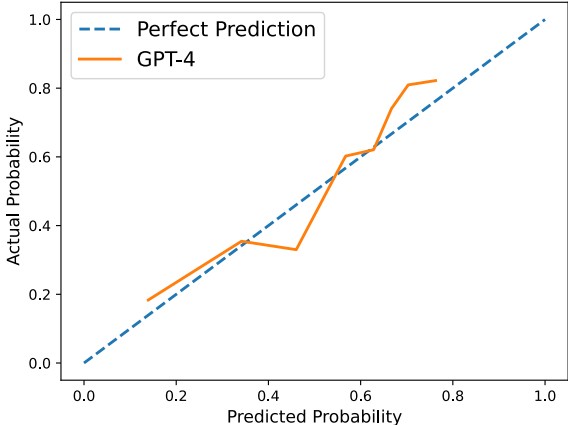

Figure 1: GPT-4 Score calibrated by Platt's method can predict probabilities (ECE=5.9%).

amples where RoBERTa was correct and GPT-4 wrong. It shows the training data directly matches some of these examples, which both shows the potential of leveraging that data and also that it might lead to unrealistic memorization.

### 4.6 Soft Classification

Our next experiment compares GPT-4 and other language models' capabilities for making probability predictions. We found GPT-4 Score can give reasonable probability predictions once calibrated (Figure 1), but SqueezeBERT's results appear to be even better, depending on how much weight one puts on identifying more certain examples vs. avoiding overconfidence. This experiment is discussed in more detail in Appendix P.

### 4.7 GPT-4 Version

The experiments in Table 2, where GPT-4 Binary achieved 66.5% accuracy, were conducted with the gpt-4-0314 version. We revisited the performance with GPT-4-0613, and found a large drop to 56.0%. Large differences from different versions been reported in other domains (Chen et al., 2023), but to our knowledge we are the first to report it in the misinformation context. We suspect that the binary prompt may cause brittleness, and explainability prompts (see Section 4.8) may be more robust by ensuring the results are based on reasoning chains rather than on the spot verdicts. We recommend this as a topic for future experimentation.

All of our experiments except for those in the following two sections were conducted with gpt-4-0314 (or the generic one, before versioning existed). Before knowing these results, Explainability used 0613, while Web Retrieval was unversioned and

run in August 2023. The Explainability performance conclusions should be taken with a grain of salt - an updated version of this paper will confirm them with 0314. The Retrieval results show huge improvement margins which are not due to versioning, so the conclusions are unaffected.

### 4.8 Explainability

GPT-4 naturally has explainability capabilities. To investigate these, we created two simple variations on the "Score" prompt, one of which consistently improves Score Zero-Shot performance in general (though see caveats in Section 4.7). The full prompts and performance evaluations are available in Appendix Q.

We performed a small sample, preliminary evaluation where an author checked each explanation for reasonableness. We found that the explanation always matched the score. Furthermore, many of the explanations for answers with the wrong label predicted were actually reasonable or partially reasonable, especially in cases with scores near 50, where a slight difference in GPT-4's evaluation vs. PolitiFact's could result in a different label. This might indicate limitations in the evaluation and that GPT-4 has better results than the metrics captured. See Appendix Q.3 for further details.

We then sampled LIAR-New examples that were Impossible[3] to investigate the model's output. In cases it got correct, besides ones it does not really know but gets lucky, it seems to either rely on general information or assume a United States context. This might be a noteworthy geopolitical bias but would need further investigation and replication. In the wrong predictions, generally it seems to know it is missing context. Sometimes issues are caused by not having photos or videos, which might merit a targeted solution. We present more details and case studies in Appendix Q.4. Overall, these results, while preliminary, show the potential of recent LLMs to help researchers understand how they are making their veracity evaluations, and in turn figure out how to improve them.

### 4.9 Web Retrieval

We conducted a preliminary experiment on the potential of web retrieval. When collecting LIAR-New, we scraped the full text of the PolitiFact articles. We provide the model with this evidence through a simple adjustment of our prompt (de-

---

[3] By V1 Possibility label—please see Appendix C

tailed in Appendix R). Because the article is guaranteed to be relevant and informative with respect to veracity, we call this approach "GPT-4 Web Oracle". We also examine a version "Web Answerless Oracle" where we remove the final veracity verdict from the PolitiFact article.

| Method | Macro F1 |
|---|---|
| GPT-4 Web Oracle | 98.6 |
| GPT-4 Web Answerless Oracle | 90.7 |
| GPT-4 Score Zero-Shot | 60.5 |
| GPT-4 Score Binary | 68.8 |

Table 9: Performance with oracle web retrieval. The improvement is massive.

We see in Table 9 a huge increase in performance. Of course, these conditions are not realistic, because we are assuming we can perfectly retrieve the PolitiFact article, which might not even exist. We note, however, that in spite of this limitation, these results are not true upper bounds, because this is a simple prompt using only a single piece of evidence. This highlights the potential of web retrieval combined with recent LLMs like GPT-4, and we suggest it is the clearest path to increasing raw performance.

## 5 Conclusions

In conclusion, this study presented a multifaceted investigation to determine if we can use GPT-4, combined with careful analysis of generalization and uncertainty, to produce more practical misinformation detection. We found, first, that GPT-4 can give superior performance for information evaluation in multiple datasets and contexts. Second, we found that GPT-4 has different failure modes from previous approaches like RoBERTa. This could help produce better results for detecting misinformation and countering its spread. Third, we investigated uncertainty quantification in this context, showing that GPT-4 can provide information linked to uncertainty and that that information can be leveraged towards significantly improved performance. Fourth, it is not easy to guarantee inputs have sufficient context for a valid judgment to be made. We provide a new dataset and analysis that starts to address this issue. However, we believe that such explorations remain important to consider in future work. Fifth and finally, we showed that both web retrieval and explainability offer promising avenues for future progress. Overall, we hope this work will provide a starting point towards both

better performance in general, and creating systems that will address key real-world factors that have so far prevented machine learning from reliably mitigating misinformation.

## 6 Limitations

We hope future work will resolve limitations of this study, including the following:

- Our analysis focused on GPT-4, which is the strongest LLM available for many tasks. We reason that if it were incapable of producing good results here, it is unlikely that weaker LLMs of the same type could do any better. Now that we know that GPT-4 gives strong performance, it is an open question to what extent these results can be replicated with other systems. For closed-source models, the largest version of PaLM 2 (Anil et al., 2023) (Unicorn), along with Bard (for which Unicorn is the foundation model) would be promising models for future evaluation. Similarly, Claude 2 would also be promising, as another leading closed-source model. For open-source models, in more recent work by authors here and collaborators (Yu et al., 2023), we showed that similar prompting of Llama-2 struggles to give good performance. However, more recent open-source models such as Zephyr[4] have generally shown superior performance to Llama-2, and are therefore also well worth evaluation.

- While our analysis shows GPT-4 improves on prior approaches, the overall results are still not as strong as one would like in many real-world applications, whether in terms of raw performance, generalization, uncertainty quantification, or many other considerations. More work is needed to produce real-world ready systems. Besides directly building on the ideas explored here, two of the most promising parallel directions are building web search systems to address examples beyond the knowledge cutoff and in general cases that are not Impossible but where GPT-4 lacks sufficient information, and building explainability systems, to create more trustworthy predictions and more graceful failure. We reported on some initial experiments in both of these directions, but there is a great deal more ground to cover in future work.

---

[4]https://huggingface.co/HuggingFaceH4/zephyr-7b-alpha

## 7 Ethical Considerations

Given the recent shifts in the spread of misinformation which has been amplified by global events such as the COVID-19 pandemic, the war in Ukraine, and the Capitol Hills riots, it is imperative that researchers improve their approach to detecting "fake news" online. The results of this study suggest a path forward to address this challenge by using the power of advanced language models like GPT-4. These systems can not only provide better performance—through careful attention to errors, uncertainty, and other real-world factors—they can also limit harms like confident mistakes and biased predictions. However, more work is needed, and we hope future research will build on this to deliver a positive real-world impact.

## Acknowledgements

This work was partially funded by the CIFAR AI Chairs Program and by the Centre for the Study of Democratic Citizenship (CSDC). The first author is supported by funding from IVADO and by the Fonds de recherche du Québec. We thank Yury Orliovsky for helping improve the quality of LIAR-New. We thank Berkeley SPAR for helping connect collaborators and start the project.

## Author Contributions

Kellin Pelrine lead the project and worked on most areas, especially the foundational ideas, experiment design, and analysis. Anne Imouza and Camille Thibault did the majority of the work to create LIAR-New, including the translation and impossibility labels. Meilina Reksoprodjo implemented most of the non-GPT-4 language models in the main comparison and contributed to writing and literature review. Caleb Gupta implemented the fuzzy clustering methods and contributed to literature review. Joel Christoph helped design and test the GPT-4 prompting and contributed to writing and literature review. Jean-François Godbout and Reihaneh Rabbany advised the project, contributing ideas, feedback, and writing.

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

## A  Related Work

Misinformation and the spread of false information have become increasingly prevalent in today's digital age (Shu et al., 2017; Sharma et al., 2019; Shu et al., 2020; Kumar et al., 2021; Shahid et al., 2022). A wide range of solutions have been proposed to tackle this problem, from fact-checking and expert-based assessments to automated machine learning approaches (Wang, 2017; Meel and Vishwakarma, 2020). Broadly, the literature can be divided into content-based and network-based approaches (Shu et al., 2017). Content-based approaches focus on analyzing the text, images, or multimedia elements

of a message to determine its veracity. Some examples of this approach include (Wang, 2017), which uses a hybrid CNN and LSTM model, and (Kaliyar et al., 2021) which employs BERT. Network-based approaches, on the other hand, examine the propagation patterns and user interactions surrounding the information (Hu et al., 2019; Long et al., 2017). Our approach is part of the content-based category.

A key challenge for practical misinformation detection is the generalizability of models across various types of datasets and domains. Several recent studies have addressed these issues from different angles. Suprem et al. (2022) investigated the generalizability of pretrained and fine-tuned fake news detectors, introducing the KMeans-Proxy method for identifying overlapping subsets of unseen data. Ni et al. (2020) employed Propensity Score Matching to select generalizable features for fake news detection, while Lee et al. (2021) proposed UNIFIEDM2, a unified model for multiple misinformation domains. Furthermore, Hoy and Koulouri (2022) explored the generalizability of popular fake news detection models and features across similar news. These studies highlight the importance of addressing generalizability challenges for effective misinformation detection. Our work here is novel in focusing on the generalizability of GPT-4 in this domain, particularly in relation to the data and other LMs.

Our approach also emphasizes uncertainty assessment, as opposed to a more rigid pursuit of overall performance metrics. The misinformation-related literature on uncertainty is very limited, lacking thorough comparative evaluations for approaches like soft classification (i.e., predicting probabilities instead of class labels) and investigations of the latest approaches, such as GPT-4 (Guacho et al., 2018; Jlifi et al., 2022; Levchuk and Shabarekh, 2017; Qu et al., 2022). Our work addresses some of the many gaps in this literature by proposing approaches to productively assess uncertainty with GPT-4, and benchmarking methods for soft classification.

## B   LIAR and CT-FAN-22 Dataset Details

### B.1   LIAR

This dataset was collected by Wang (2017) from the PolitiFact.com API and includes 12.8K human-labeled short political statements. For the degree of truthfulness, there are six labels: Pants-on-Fire, False, Barely-true, Half-true, Mostly-true,

and True. The labels are fairly balanced, except for the Pants-on-Fire label, which has roughly half as many examples as the others. The dataset provides an approximately 80-10-10 train-val-test split. Table 10 shows the basic statistics.

In addition to statements, each example includes metadata: the statement topic, and the speaker's name, job, state, party, and total "credit history". The latter refers to the counts of statements of each label by the speaker in the training set, excluding "True". Prior research (Hu et al., 2019; Long et al., 2017) has found that most of the metadata did not give significant performance benefits, but the credit history improved performance significantly. However, because statements for fact-checking by PolitiFact are not sampled randomly, using this metadata may cause models to learn sampling bias that can aid classification in this dataset, but will not generalize well to the real world. For instance, a speaker might have controversial false statements which are fact-checked by PolitiFact, leading to a negative credit history, when they make many other uncontroversial true statements that are not fact-checked by PolitiFact and consequently not part of the dataset. This contrasts with content-based classification: both may fail to classify unseen topics or speakers, but speaker-based classification (particularly when based on a biased sample) may fail even for speakers seen in the training data. In addition, the credit history may be impossible to use for classifying statements from non-famous individuals (such as newly-created social media bot accounts) and introduce ethical issues. For these reasons, we exclude metadata and focus exclusively on the statements themselves. This is also a common experimental setup in the literature (Shu et al., 2017; Sharma et al., 2019; Shu et al., 2020; Kumar et al., 2021; Shahid et al., 2022).

Both 6-way classification with the full set of labels, and binary classification with the labels collapsed into two, are common approaches to this dataset in the literature. We focus on the binary case for more lucid error and generalization analyses, but also report 6-way classification results. For the binary experiments, we dichotomized the categories by splitting the labels in the middle—i.e., mapping Pants-on-Fire, False, and Barely-true to "False" and Half-true, Mostly-true, and True to "True". While not the only option, this is the most typical binarization used in the literature (Qu et al., 2022; Orsini et al., 2022; Khan et al., 2021; Flores

| | Pants-fire | False | Barely-true | Half-true | Mostly-true | True | Total |
|---|---|---|---|---|---|---|---|
| **Train** | 839 | 1995 | 1654 | 2114 | 1962 | 1676 | 10240 |
| **Val** | 116 | 263 | 237 | 248 | 251 | 169 | 1284 |
| **Test** | 92 | 249 | 212 | 265 | 241 | 208 | 1267 |

Table 10: LIAR dataset statistics.

and Hao, 2022).

Note that it is also common for studies to code the LIAR dataset in a slightly different way than the original splits—e.g., testing on both the validation and test sets together. Although these may not be perfectly comparable, we include this classification in consideration for the state-of-the-art method to compare with our results, providing a more robust benchmarking for our own approach. On the other hand, in addition to studies using the metadata as discussed above, we exclude from comparison some studies like Pryzant et al. (2023), which evaluate their approach on a very different subset of the LIAR dataset, as well as ones which use a different binarization, such as Dong et al. (2022); Panda and Levitan (2022). To our knowledge, the resulting state-of-the-art approaches are Orsini et al. (2022) and Khan et al. (2021) tied at 62% accuracy for binary classification, and Flores and Hao (2022) with 29.4% for 6-way classification.

## B.2 CT-FAN-22

Köhler et al. (2022) derived this dataset from 20 different fact-checking websites. The labels were converted from each website's own system to four: False, Partially False, True, and Other. There is both a large English corpus, and a test-set-only German corpus (collected separately, not translated).

| | | False | Partially False | True | Other | Total |
|---|---|---|---|---|---|---|
| **English** | Train | 465 | 217 | 142 | 76 | 900 |
| | Val | 113 | 141 | 69 | 41 | 364 |
| | Test | 315 | 97 | 243 | 55 | 612 |
| **German** | Test | 191 | 97 | 243 | 55 | 586 |

Table 11: CT-FAN-22 dataset statistics.

The "Other" label could be problematic because it cannot guarantee something for which veracity is unverifiable; it only reflects that an entry has not been verified in this dataset. Furthermore, it is not a category which can be synchronized with other datasets, impeding transfer analyses. In one of our experiments, we maintain this category for comparative purposes only, and directly mark every prediction by our models on examples with this ground truth label as wrong. This leads to a maximally stringent evaluation favoring previous approaches. In another set of experiments, we re-move this category and evaluate on the remaining examples. While we believe going beyond simplistic labeling and metrics is critical to capturing more real-world information and producing more generalizable systems, we suggest that this "Other" label is not well-defined and should be excluded from future datasets, either in testing or directly in the dataset construction.

The state-of-the-art approaches are reported by Köhler et al. (2022) from a competition run on this dataset. For English, the strongest performance is by Taboubi et al. (2022) with 54.7% accuracy and 33.9% macro F1. For the more challenging German corpus, the strongest approach was found by Tran and Kruschwitz (2022), with 42.7% accuracy and 29.0% macro F1. We note that these approaches used both the article title and text that the dataset provides, while for our experiments, we only used the article text because of limited time and resources.

## C Possible vs. Impossible Example Labeling

We created two versions of LIAR-New Possibility labels. We start by describing the first version and its limitations that led us to revisit the labeling. We then discuss the second, current version. The second version is now used in all experiments unless otherwise noted.

### C.1 V1 Possibility labels

We began with the following definition:

**Definition 2.** *A statement is considered "Impossible" with respect to veracity judgment if it meets one of the following criteria: 1. it does not contain any statement of verifiable fact (e.g., the phrase "On taxes" in isolation has no claim that could be evaluated) or 2. the fact that the statement references cannot be verified without unavailable context (e.g., "they voted for it" requires knowing who "they" are and what "it" refers to). Conversely, a statement is considered "Possible" if it can be confirmed or disproven with external knowledge, but does not require additional unavailable context (e.g., "the sun orbits the earth" may require astron-*

*omy or physics knowledge to disprove, but does not require additional context).*

In this definition, there are only two categories, Possible and Impossible. Two authors (Anne Imouza and Camille Thibault) each annotated half of LIAR-New according to this definition. This yielded 892 Possible and 1063 Impossible examples.

Each annotating author then annotated 100 examples from the other's set, and we evaluated disagreement. We found that disagreement on this combined set of 200 examples was significant, with Cohen Kappa 0.312 and 72/200 cases disagreeing.

The annotating authors discussed each of these 200 disagreement cases and resolved the disagreement. We did not observe a clear pattern such as all disagreements being resolved to one label or the other; it varied case by case. Consequently, although this is a challenging and potentially subjective labeling task, to improve quality as much as possible we decided it would be beneficial to revise all the labels. Based on the annotating authors' observations and group discussion, we decided to add a third category, leading to Version 2 discussed below.

We note however that the experimental conclusions did not change substantially between the two versions, suggesting the original version was unbiased, if noisy.

### C.2 V2 Possibility labels

The following definitions are the full versions of the ones we revised to, including examples (which were agreed upon in advance of the labeling) omitted in the main text:

**Definition 3.** *Possible: the statement's claim is clear without any additional context (e.g., a statement "the earth is flat"), or any missing context does not make it difficult to evaluate the claim ("at the press conference, Senator X said the earth is flat" - we may not know which press conference this refers to, but we can give a strong veracity evaluation just examining if Senator X has said the earth is flat at all, in a particular context, or frequently).*

*Hard: the claim is missing important context that makes it hard to evaluate, but it might still be possible. For example, "Senator X is against the $15 million cut." We might be able to trace the amount and the senator's name and figure out what this refers, and give a practical evaluation likely beginning with "Assuming the cut here refers*

*to..." But it's challenging and might be impossible in some cases, for example if it's not possible to figure out which cut this refers to.*

*Impossible: there is missing context that cannot be resolved ("The senator said the earth is flat" - without a name, this could be any senator throughout history, so it's not possible to give a valid or productive veracity evaluation). Or, the statement contains no claim for which veracity could be evaluated regardless how much context is available (for example, a statement "on the shape of the earth" contains no claim).*

The 3-way labels here provide more information and can make it easier to handle edge cases than the rigid 2-way labels. We also recruited a third person (Yury Orliovsky) to provide a third set of labels to further reduce noise.

In this round, each annotator annotated every example. The resultant counts of labels are shown in Table 12.

We see that the dominant cases are either agreement or a relatively mild form of disagreement between Possible vs. Hard or Impossible vs. Hard. We consider this type of disagreement to be within acceptable bounds, since by definition the Hard class might in fact be either Possible or Impossible. Therefore, for the final dataset, we resolved these disagreements by taking the majority vote.

Nonetheless, there are also 342 cases where at least one annotator marked Possible and another marked Impossible. In these cases, the labels are clearly conflicting. To resolve these disagreements, we split them into two categories: 127 "maximal" disagreement cases where each label is represented, and 215 "nonmaximal" where two annotators marked one label and one marked the other.

For the maximal disagreements, the annotators discussed each example live and resolved the disagreement to produce a final label. For the nonmaximal disagreements, the minority vote author first went through each case individually to assess if they had mislabeled it on their end. Finally, for cases where the minority vote author in their individual assessment still disagreed with the majority, all the annotators discussed live to produce a final label.

This yielded 927 Possible, 581 Hard, and 449 Impossible examples. We note that in the experiments here we only examine these three categories, however, potentially the raw labels from the individual annotators could be used to form a finer-grained

scale. For example, a case all three annotators labeled Hard might be harder than a case where two labeled Hard and one labeled Possible. However, caution would be needed here as it could result in noise.

We make both the raw and final labels available for future work through our GitHub.

# D In-context learning with automatically chosen demonstration

## D.1 Prompts

We tested several prompts in which GPT-4 is provided an example chosen automatically from the training set. These were motivated by the observation that GPT-4 and RoBERTa make different errors in a way that is tied to embedding similarity with training set examples (Section 4.5). Consequently, by providing the most similar training set example, we hoped to provide both a demonstration and potentially a direct source of information relevant to the target example. Related procedures to retrieve semantically similar examples have shown performance benefits in other contexts (Luo et al., 2023).

In the below prompts, CLOSEST_TRAIN_LABEL refers to the label of the closest training example according to Ada-002 embedding distance, uniformly mapped onto the 0-100 range. CLOSEST_TRAIN_TEXT refers to the corresponding training example input statement.

Version 1:

> "role": "user", "content": 'Rate the truthfulness of the following statement: "STATEMENT"
>
> Here is the rating of a similar statement. The rating is accurate, but the statement may or may not be relevant to this one. Similar statement: "CLOSEST_TRAIN_TEXT" Rating: CLOSEST_TRAIN_LABEL
>
> Provide a score for the original statement from 0 to 100, where 0 represents definitively false and 100 represents definitively true. Do not provide any explanations, only respond with the numerical score.'

Version 2:

> "role": "user", "content": 'Here is a similar statement that is CLOSEST_TRAIN_LABEL: "CLOSEST_TRAIN_TEXT" Now rate the truthfulness of the following statement: "STATEMENT" Provide a score from 0 to 100, where 0 represents definitively false and 100 represents definitively true. Do not provide any explanations, only respond with the numerical score. '

| Labels | Count |
| --- | --- |
| ppp | 270 |
| hhp | 157 |
| hii | 129 |
| hpp | 69 |
| php | 62 |
| ppi | 59 |
| iii | 52 |
| hhh | 41 |
| hhi | 39 |
| pph | 34 |
| pii | 31 |
| phh | 25 |
| phi | 22 |
| hpi | 19 |
| hih | 16 |
| hip | 16 |
| ihp | 15 |
| iip | 14 |
| pip | 14 |
| ipp | 11 |
| hph | 10 |
| iih | 9 |
| ihi | 7 |
| ihh | 7 |
| pih | 4 |
| ipi | 3 |
| iph | 2 |

Table 12: Counts of each label combination, with order preserved from the annotators. "ppp", for example, indicates all three annotators marked the example "Possible".

Finally, Version 3 is a meta-prompt where we set a condition to select between Version 2 and the original (no demonstration) Score Zero-Shot prompt. Given the example we want to classify, as in previous versions, we find the most similar training set example. Then if this train-test example pair is among the top 10% most similar train-test example pairs (i.e., the similarity between the target, test set example and it's most similar training set example is in the top 10% similarity of all such pairs), we use the Version 2 prompt. Otherwise, we use the Score Zero-Shot prompt. The motivating intuition here is that if we have a very similar training set example, then we try to take advantage of that. In contrast, if none of the training set examples are very similar, then we hypothesize they may not be relevant and therefore including them might confuse the model, so we revert to the Zero-Shot prompt.

### D.2 Results

We report results in Table 13. We see that version 1 seriously damages performance. Version 2 is more reasonable but still slightly worse than Zero-Shot. Version 3 is better than Zero-Shot, but the difference is marginal and unlikely to be worth the complications and potential detriment to generalization of relying on a limited, static training set.

| Method | Accuracy |
|---:|:---:|
| Version 1 | 57.2 |
| Version 2 | 64.2 |
| Version 3 | 65.2 |
| GPT-4 Score Zero-Shot | 64.9 |

Table 13: Results of 3 versions of in-context learning with an automatically chosen demonstration. Version 3 improves performance compared to Zero-Shot, but only marginally, so the benefit is likely not worth implementation complications and potential impact on generalization.

We note that this is only using a single example—perhaps more are necessary to minimize the impact of noise. Also, there could be many other ways to do the prompting, for instance maybe this does not synergize well with the Score prompt but would work better for Binary. So we cannot rule out some related approach doing better. But we believe other ways to provide the model additional information, especially web retrieval (see Section 4.9), are much more promising.

## E  Small Language Models

We fine-tuned a selection of pre-trained SLMs from HuggingFace. Due to limited computational resources, we restricted our analysis to models that were runnable on one Tesla T4 GPU. We examined BERT (base) (Devlin et al., 2018), ConvBERT (Jiang et al., 2020), DeBERTa and DeBERTa-v3 (base) (He et al., 2020), LUKE (Yamada et al., 2020), RoBERTa (base) (Liu et al., 2019), Squeeze-BERT (Iandola et al., 2020) and XLM-RoBERTa (Conneau et al., 2019).

A small hyperparameter search was performed on the given models, however, we rapidly found that this was not significantly increasing performance and began showing signs of overfitting. Hence, for this purpose, all the reported values in Table 2 as well as Table 20 are based on default configurations for pre-training BERT-like models, as stated in the original BERT paper. All the models were finetuned for 2 epochs, using the base embeddings with the learning rate 5e-05 using the AdamW optimizer.

## F  Fuzzy Clustering

K-Means Fuzzy Clustering has shown promising results for misinformation detection in some recent studies (Chen and Lai, 2022; Raj and Meel, 2021). This method initializes centroids in a pseudo-random fashion, then each datapoint is assigned a probabilistic membership to each cluster proportional to the distance between the data point and centroid. Following this, the centroids themselves are moved relative to the averages of their data points. Memberships and centroids are updated repeatedly until convergence. The final membership values can be converted to a class prediction with an argmax or an additional classifier.

We tested fuzzy clustering using three types of word embeddings with various levels of global contextualization: Word2Vec, GloVE, and BERT. The models were trained using adaptations to the SciKit-Fuzzy cluster cmeans library.

Unfortunately, in our experiments (Tables 2 and 20), this approach did not give good results, so we did not investigate it further.

## G  GPT-3.5

### G.1  Base Performance

We prompt GPT-3.5 in the same way as GPT-4, and evaluate its performance in Table 14.

| Dataset | Method | Accuracy |
|---------|--------|----------|
| LIAR | GPT-3.5 Score Zero-Shot | 67.3 |
| LIAR | GPT-3.5 Binary | 53.6 |
| LIAR-New | GPT-3.5 Score Zero-Shot | 61.1 |
| LIAR-New | GPT-3.5 Binary | 55.7 |
| LIAR-New | GPT-3.5 Binary Uncertainty-Enabled | 67.7 |

Table 14: Results with GPT-3.5.

We note that GPT 3.5 performs surprisingly well on LIAR, better than GPT-4 in the same Score Zero-Shot setup. However, there are some caveats. First, its score is already maximized at the zero-shot binarization threshold of 50, meaning it cannot achieve higher performance through tuning on the validation set like GPT-4 can. The margins are small, however, so this may not have a big impact. More significantly, the performance seems brittle and does not extend to other prompts or datasets.

### G.2 Fine-Tuning

We conduct a preliminary experiment on fine-tuning using the following prompt:

> "role": "user", "content": 'Rate the truthfulness of the following statement: "STATEMENT" Provide a score from 0 to 100, where 0 represents definitively false and 100 represents definitively true.'

This is the same as the usual Score prompt, except we remove the last sentence "Do not provide any explanations, only respond with the numerical score." We hypothesized that the finetuning should already be sufficient to get it to respond with only the numerical score—and in practice it does, 100% of the time.

For the output side of the fine-tuning, we map the original 6-way LIAR labels uniformly to 0-100 scores, i.e., [0,20,40,60,80,100]. Then we fine-tune with default settings, which resulted in 2 epochs of fine-tuning.

On the test set, this approach achieved 57.2% accuracy. Considering the corresponding untuned approach (Score Zero-Shot) achieved 67.3% accuracy, the fine-tuning was not successful. We hypothesize that finetuning to Impossible examples might be particularly damaging. In addition, the Score prompting may not be well suited for this since the uniform mapping may not reflect the true variation in the labels. So there is considerable room for more experimentation here that might yield a better result. We recommend this, along with testing GPT-4 fine-tuning that OpenAI has said it plans to release in the future, as areas for future work.

## H Evaluation Metrics

To measure the performance of our misinformation detection approaches in hard classification settings we mainly report accuracy and weighted F1 score. For LIAR-New, where the classes are very imbalanced, we report accuracy and macro F1 score, which gives a more stringent assessment. Meanwhile, for soft classification, we provide reliability diagrams and measure Expected Calibration Error (ECE), which are standard ways to measure performance in this context (Abdar et al., 2021). Both were applied using 10 bins with quantile scaling, i.e., the bins are scaled to have the same number of examples in each.

## I GPT-4 Cost Evaluation

We report in Table 15 approximate API costs of some of the main experiments run. These numbers can vary between runs of the model (due to randomness in the completion tokens) but should provide reasonable estimates. This is based on current pricing for GPT-4 (8K context window) as of October 2023, i.e., USD 0.03/1K input tokens and 0.06/1K output tokens.

We also note that the finetuning of GPT-3.5 on the LIAR train set comprised 609k tokens costing 4.9 USD.

Overall, we see that there is a very large range depending on different data and prompts. We note that if an explanation is not needed, prompts that avoid it can be quite effective in reducing cost in cases where the input is not long (e.g., LIAR and LIAR-New). Furthermore, the overall budget needed for these experiments is not trivial, especially considering that not all experiments are included here and many need multiple runs (e.g., when evaluating temperature in the following section, doing exploratory analysis, debugging, etc.). In the short-term, thoughtful prompting and choice of experiments to target can help keep costs down when budget is limited. In the long-term, cheaper models with comparable performance will hopefully (and quite likely) be found or created, which will be a significant advantage for both research and real-world deployment.

| Dataset | Prompt | Input Tokens | Output Tokens | Total Cost (USD) |
|---|---|---|---|---|
| LIAR | Score | 100K | 3K | 3.19 |
| LIAR | Binary | 93K | 2K | 2.88 |
| LIAR | Explain-then-Score | 142K | 314K | 23.1 |
| LIAR | Score-then-Explain | 104K | 118K | 10.21 |
| LIAR-New | Score | 154K | 2K | 4.73 |
| LIAR-New | Binary | 140K | 2K | 4.32 |
| LIAR-New | Binary Uncertainty-Enabled | 205K | 4K | 6.37 |
| LIAR-New | Explain-then-Score | 216K | 464K | 34.35 |
| LIAR-New | Web Oracle | 1.8M | 2K | 53.07 |
| LIAR-New | Web Answerless Oracle | 1.7M | 2K | 51.07 |
| CT-FAN English | Score | 725K | 1K | 21.79 |
| CT-FAN English | Explain-then-Score | 691K | 197K | 32.57 |
| CT-FAN German | Score | 956K | 1K | 28.72 |

Table 15: GPT-4 API costs.

## J Temperature and Variation

We examine here the effect of varying the sampling temperature from 0.5, used in the main experiments, to 0.0 or 1.0. There are two outcomes to consider: first, the effect on overall performance. Second, the variation of GPT-4's responses for each example over multiple runs. We test these by running GPT-4 10 times on the test data at each temperature level, in the binary setting.

In Table 16 we report first the mean accuracy and its standard deviation over 10 runs. Then we report the number of non-numeric answers, e.g., when GPT-4 refuses to answer or returns an explanation instead of following the directive to only return a score 0-100. Note that for the accuracy calculation, we replaced these with a random number uniformly drawn between 0 and 100, while in the subsequent metrics we exclude these examples. For each example, we then compute the standard deviation of the scores, and report the mean and maximum of that value over all the examples in the dataset (Mean SD and Max SD). Finally, we look at peak-to-peak distance, i.e., the difference between the largest and smallest GPT-4 scores for each example over the 10 runs. We report the maximum (Max PtP) and the number of examples for which the PtP distance is over 50 (# Large PtP).

We find that lower temperature generally gives better results, with both a small increase in accuracy and a reduced variation amongst the responses. The mean standard deviation between runs with 0.0 temperature is quite reasonable at 2.2, indicating the responses are generally consistent. We do however note that a small number of examples still have

| Temp. | Accuracy | # Non-numeric | SD Mean | SD Max | PtP Max | PtP # Large |
|---|---|---|---|---|---|---|
| 0.0 | $67.9 \pm 0.3$ | 95 | 2.2 | 51.6 | 100 | 5 |
| 0.5 | $67.4 \pm 0.7$ | 108 | 5.3 | 48.3 | 100 | 11 |
| 1.0 | $66.5 \pm 0.8$ | 129 | 7.7 | 40.7 | 100 | 40 |

Table 16: Temperature vs. accuracy and several measures of variation in the responses. In general, lower temperature gives better results.

a very large variation in responses—regardless of the temperature, there is always an example with the maximal 100 peak-to-peak distance, and even with 0.0 temperature there are 5 examples with over 50 peak-to-peak distance. This might be troublesome for user-facing systems built on this, as occasionally it can give very different answers to different users querying the veracity of the same statement, or one user querying multiple times, which could cause some confusion or distrust.

Consequently, future work that addresses this limitation would be helpful. This might be done with better prompts or with techniques such as multi-agent debate (Du et al., 2023; Liang et al., 2023), or possibly giving the system more context and evidence.

We also note that the maximum standard deviation shows a counterintuitive result, increasing as temperature decreases. This might be noise, or it might be due to lower temperature following the instruction to respond with a number more consistently and still giving an answer in cases where it is more confused. We found that of the 95 examples where at least one run at 0.0 temperature returned a non-numeric answer, all but one were contained

within the 129 examples where a run at 1.0 temperature returned a non-numeric answer. Furthermore, excluding the full 129+1 examples increased performance to 69.6 ± 0.2, reduced Mean SD to 2.1, and removed one of the 5 large PtP cases. It did not change the maximum standard deviation or peak-to-peak distance, but nonetheless, this might suggest that higher temperature has an edge in realizing cases it cannot answer properly. Further investigation is needed to determine if an adjusted prompt, which directly allows the model to refuse to answer, would have a positive effect here.

Finally, we also considered the potential effect of changing the temperature, as well as repeated runs in general, on the optimal threshold. We evaluated all possible thresholds to determine the oracle threshold that would give the best performance at each temperature. Results are shown in Table 17. We see that the original threshold found on the validation set (with 0.5 temperature) was a bit high for the test set, especially at 0.0 temperature. However, the difference in accuracy between this threshold and the oracle one is quite small. Thus, it does not appear that temperature has a large practical effect on the optimal threshold.

| Temp. | Original Thre. | Accu. | Oracle Thre. | Accu. |
|---|---|---|---|---|
| 0.0 | 71 | 67.9 | 62 | 68.1 |
| 0.5 | 71 | 67.4 | 66 | 67.7 |
| 1.0 | 71 | 66.5 | 67 | 67.5 |

Table 17: Temperature vs. performance with the original threshold from one run on the validation set (71), and the oracle threshold that maximizes performance. The original threshold is somewhat high, but the difference in accuracy is low.

## K  LIAR Possible vs. Impossible Examples

In addition to the larger scale analysis on LIAR-New, we also examined the Possible vs. Impossible examples in LIAR (i.e., using the ones that GPT-4 found for the Impossible case, as discussed in Section 3.1). We restrict the test set to only one or the other category, and compare results with GPT-4 and RoBERTa-large. First, in Table 18 we see that performance improves slightly (compared to Table 2) when restricting to the Possible examples, which confirms our initial intuition.

Next, for the Impossible examples, we see the results of flipping a coin vs. RoBERTa-large. RoBERTa outperforms random classifications here,

which means it may be learning information or a bias which helps on this data but will not generalize. On the other hand, if we look at RoBERTa's soft classification of these examples, the results are more encouraging: the mean absolute deviation from 0.5 (which would match "can't be determined one way or another") is 0.232 in the Impossible case, compared to .338 in the Possible case. This suggests more certainty overall on Possible examples, which is the direction one would want, even if it is not sufficient to achieve the desired outcome.

| | Method | Accuracy | F1 |
|---|---|---|---|
| **Possible** | GPT-4 | 69.4 | 69.1 |
| | RoBERTa-large | 63.8 | 62.3 |
| **Impossible** | 50-50 Random | 50.0 | 50.2 |
| | RoBERTa-large | 59.4 | 58.4 |

Table 18: Possible vs. Impossible examples. RoBERTa-large beats random on Impossible ones – which is might not generalize.

Finally, we further tested what happens to the GPT-4 classification threshold when these examples are included, as in Table 2, by assigning them a random prediction. The threshold increased from the optimal 71 to a mean over 100 runs (re-randomizing the Impossible predictions) of 72.31. Although not a large change, this does illustrate concretely how including these examples can bias the predictions.

Overall, these results confirm that noise in the data can affect downstream results. RoBERTa-large can somewhat account for them, but not nearly to the extent of GPT-4. Since it can detect many Impossible examples, GPT-4 has a robustness advantage in this area, and might also be used for cleaning the data if approaches like RoBERTa-large are needed.

## L  Investigating Errors in Relation to the Training Data

We embedded all examples with OpenAI's Ada-002 model (Greene et al., 2022). This is a document embedding model which is suitable for evaluating their semantic similarity. For each example in the test set, we find the most similar statement in the training set by calculating the cosine distance in Ada-002 embeddings between the statements. We find the training set statement that minimizes the distance to each test set statement. Finally, we compare the minimum distances averaged over all

the test set examples.

For the cases where GPT-4 was correct and RoBERTa-L wrong, the average distance is 0.127, while for the reverse, the average is 0.116. The difference is statistically significant (p=0.0005). This suggests RoBERTa is using similar examples in the training data to make correct predictions for cases GPT-4 gets wrong, while GPT-4 must be using the information it learned during its own training process to correctly predict cases that do not relate well to the examples RoBERTa sees in the training set. We publish on our Github these examples and the associated cosine distances so that further analysis on the generalizability of RoBERTa's predictions can be conducted. If replicable in other contexts, these results suggest GPT-4 or similar models could benefit significantly from effective finetuning or few-shot setups in this domain, if they can be optimized beyond the failed attempts we made here (Appendix G and Appendix D). In other words, if GPT-4 could correctly classify all the examples RoBERTa does without compromising its current correct examples, there would be a nearly 10 percentage point improvement. However, the optimal methods to achieve that—without compromising generalization—remain an open question.

## M  LIAR Error Analysis Case Study

We present here some examples that RoBERTa-L Binary gets correct and GPT-4 Score Zero-Shot gets wrong, along with their closest training set example by Ada-002 embedding cosine distance. Specifically, we randomly sampled 50 examples for qualitative analysis, and here we print the five with the lowest cosine distance, the five in the middle, and the five with the highest cosine distance among the 50. From the closest five, we see that a number of examples in the LIAR test data are near-replicas of an example in the training data. Some others are very similar. On the one hand, this means that the training data is helpful to classify some examples, explaining why RoBERTa-L gets some right that GPT-4 gets wrong, and suggesting a potential benefit if GPT-4 could ingest the training data directly. On the other hand, this also means some examples might be artificially easy for models that have seen the training data, and can be classified by memorization rather than learning generalizable patterns.

## N  LIAR 6-Way Classification

In Table 20, we present results for 6-way classification—i.e., using the original LIAR dataset labels, without dichotomizing the ratings. In this case, DeBERTa performs best. GPT-4 Zero-Shot is not far behind, but this suggests that more information is needed to match the finer-grained labels here, either from the training set or by tuning the GPT-4 classification thresholds on the validation set. Performance is very low overall, but this matches the literature, and in the case of DeBERTa, exceeds it (Flores and Hao, 2022). It is important to note that, as shown by the low metrics across the board, this task is extremely difficult. Besides current approaches just being insufficiently capable in general, this could potentially occur if there were a lack of consistency in the fine-grained labeling, so further validation of the labels could be a valuable area for future work.

## O  PaLM 2 Bison

In Table 21, we examine the performance of the Score prompt with PaLM 2 Bison (Anil et al., 2023). We report accuracy for LIAR and Macro F1 for CT-FAN (the most used metrics for those datasets).

We see that Bison performs OK, if not impressively, on LIAR. It is significantly worse than GPT-4 and other approaches on CT-FAN. In addition, unlike with GPT-4, we did not find any significant performance gains from tuning the score threshold. Nonetheless, the results on LIAR do suggest reasonable misinformation detection is no artifact of GPT-4 alone among recent LLMs. and similarly capable models might give similarly strong results.

Bison is the second largest PaLM 2 model. The largest, Unicorn, is not publicly accessible in a direct way as of this writing. Unicorn can be used indirectly as it powers Google Bard, but Bard is not accessible in the authors' country (Canada), and to our knowledge does not offer an API anywhere. Consequently, we were unfortunately not able to test test the most powerful version of PaLM 2. If accessibility issues can be resolved, we believe that would be a worthwhile evaluation for future work.

## P  Soft Classification

Here we evaluate the capabilities of GPT-4 and other language models for making probability predictions. We first discuss the calibration method we used, then the results.

|  | Test Set Statement | Closest Train Statement |
|---|---|---|
| **Closest 5** | Weve doubled the production of clean energy. | Weve doubled our use of renewable energy. |
| | Says the National Labor Relations Board told Boeing that it cant build a factory in a non-union state. | Says the National Labor Relations Board told Boeing that it couldnt build a factory in South Carolina because South Carolina is a right-to-work state. |
| | Under Barack Obama, the U.S. now has the lowest workforce since (President Jimmy) Carter. | Under Barack Obama, the fewest number of adults are working since Jimmy Carters presidency. |
| | Says (Clinton) called President Assad a reformer. She called Assad a different kind of leader. | Says Hillary Clinton defended Syrias President Assad as a possible reformer at the start of that countrys civil war. |
| | Gov. McDonnells proposed budget is cutting public education. | Says Gov. Bob McDonnells budget would cut pre-kindergarten programs. |
| **Middle 5** | PolitiFact Texas says Congressman Edwards attacks on Bill Flores are false. | Says PolitiFact has ruled that Ed Gillespies ads are false and misleading. |
| | One in five American households have nobody under the same roof thats got a job. | We have 25 million Americans out of work. |
| | Says Bill Pascrell voted to eliminate all estate taxes for billionaires, and I voted no. | Says Congressman Bill Pascrell voted to remove the public option from the Affordable Health Care Act. |
| | Of all cities in the United States with more than 100,000 people, Providence is the 183rd safest. | [When I was mayor] Providence was one of the five renaissance cities of America, according to USA Today in those days. Money magazine said it was the fifth best city to live in in America. In addition to that, All Cities Almanac said in 1994, I believe, that it was the safest city in America. |
| | Says critics who say he cut Medicaid are wrong; his budget added $1.2 billion to the program | Says he opposed $716 billion cut to Medicare. |
| **Furthest 5** | Says Republican Rep. Charlie Dent wants to kick the Freedom Caucus out of the Republican conference for voting our conscience. | Says Congressman Eric Cantor wants to eliminate Social Security. |
| | Active duty males in the military are twice as likely to develop prostate cancer than their civilian counterparts. | One in five women in the military are receiving unwanted sexual contact, as are 3.3 percent of men. |
| | We have 41,000 people right now on the waitlist for financial-based assistance for our tech colleges and universities, most of them for tech schools. | The 2013-15 state budget backed by Gov. Scott Walker and Republican lawmakers left our technical colleges funded at 1989 levels. |
| | Ninety percent of the people in Arkansas still love Bill Clinton. | 59 percent of Americans today believe that Barack Obama is still a Muslim. |
| | When people enter the service, theres not a mental health evaluation. | Seventy-five percent of the young adults in this country are not mentally or physically fit to serve. |

Table 19: Samples of LIAR examples that RoBERTa-L Binary classified correctly and GPT-4 Score Zero-Shot classified incorrectly, grouped by semantic similarity according to Ada-002 embeddings. The closest ones are near identical, the middle ones are similar, the furthest ones are only tangentially related.

## P.1 Calibration

For soft classification (probabilities), we first use the scores our models provide out-of-the-box. In the case of GPT-4, we use the raw output it returns (divided by 100 to put it between 0 and 1). For the other language models, we take the output of the final softmax, right before the argmax converts this value to a single class prediction. These approaches have the advantage of simplicity and not introducing potential confounding variables.

Note, however, that the literature suggests deep learning models are often over-confident and their out-of-the-box probabilities may be of low quality (Moon et al., 2020). Therefore, we apply Platt's method (Jiang et al., 2011) to calibrate. This involves fitting a logistic regression model to the model's output probabilities. We do this on the validation set. This will in theory help correct for overconfidence and lead to better performance. For evaluation metrics, please see Appendix H.

Aside from GPT-4 which is not trained, all the models for this section were trained in the binary classification setting of Table 2.

## P.2 Results

Results are shown in Figure 2 and the following figures. We first observe that Platt's method generally improves the quality of the predictions. We next see that GPT-4 predictions are not very well calibrated out-of-the-box, but after Platt they show a decent relationship between prediction and actual probability, indicating this approach has potential to give more informative predictions. We suspect

| Method | Accuracy | F1 |
|---|---|---|
| SOTA (2022) | 29.4 | - |
| GPT-4 Zero-Shot | 28.1 | 25.5 |
| BERT | 28.1 | 26.4 |
| ConvBERT | 27.9 | 27.2 |
| DeBERTa | **29.9** | **29.2** |
| DeBERTa-V3 | 28.9 | 27.5 |
| LUKE | 28.3 | 27.0 |
| RoBERTa | 26.5 | 22.1 |
| SqueezeBERT | 26.6 | 25.3 |
| XLMRoBERTA | 20.9 | 07.0 |
| Fuzzy (Word2Vec) | 22.3 | 22.8 |
| Fuzzy (BERT) | 23.1 | 23.2 |
| Fuzzy (GloVE) | 21.5 | 21.1 |

Table 20: 6-Way Classification Results (percentages)

| Dataset | Method | Result |
|---|---|---|
| LIAR | PaLM 2 Bison | 62.4 |
| LIAR | GPT-4 Score Zero-Shot | 64.9 |
| CT-FAN English | PaLM 2 Bison | 21.3 |
| CT-FAN English | GPT-4 Score Zero-Shot | 42.8 |
| CT-FAN German | PaLM 2 Bison | 22.3 |
| CT-FAN German | GPT-4 Score Zero-Shot | 38.7 |

Table 21: Comparison with PaLM 2 Bison. GPT-4 performs better, but Bison does give passsable results on LIAR. Note that this is only the second largest PaLM 2 model; the largest was not accessible to the authors as of this writing.

that this could be improved even further with a better prompt.

More surprisingly, we find that superior performance on hard classification does not translate to superior performance on soft classification. The best soft classification model is SqueezeBERT (after calibration), which gave thoroughly unremarkable performance in the hard classification experiments. We believe this warrants further investigation for two reasons. On the one hand, this could lead to an effective, nuanced misinformation mitigation tool leveraging the best-performing approaches here like SqueezeBERT. On the other hand, understanding why different approaches perform better or worse here for soft classification despite their hard classification performance could lead to better soft classification in general.

Finally, we see that although after calibration the different approaches do overall follow the perfect prediction line, especially SqueezeBERT, they do not give many predictions with very high certainty

(close to 0 or 1). This can be an issue because we still want systems that make strong predictions in clear cut cases. While this is likely better than making numerous confident but wrong predictions, as one gets with standard hard classification, it suggests a line of future work aimed at fully realizing the potential of soft classification approaches.

## Q Explainability

In many experiments, we have examined prompts that explicitly discourage explanations. This gives advantages in speed and reduced cost (see Appendix I), and such outputs are easier to evaluate quantitatively. However, in the research and development context, explanations could provide insights into how the system is making evaluations and how we could improve it. In a deployment context, they could enable a user to see if its evaluations seem complete or if something is missing, leading to improved trust and more graceful failure. In the following subsections we go into more detail on our explainability experiments.

### Q.1 Prompts

We considered two prompts. The first, "Score-then-Explain" is very close to the original "Score" prompt. We simply replace the last line in the original, which discourages explanations, with a line requesting them.

> "role": "user", "content": 'Rate the truthfulness of the following statement: "STATEMENT" Provide a score from 0 to 100, where 0 represents definitively false and 100 represents definitively true. Then write a vertical bar "|", followed by explaining your reasoning for the score.'

The second prompt, "Explain-then-Score", asks the model to write an explanation before deciding on its score. The intuition here is that if we ask for a score first and lock that in, then the explanation will likely be written to match the score. But if we ask for the explanation first, the model will have a chance to reason further about the input as it writes the explanation, which might lead to a more thoughtful and accurate score. This approach was directly inspired by (Tian et al., 2023), and has connections to general chain-of-thought (Wei et al., 2022) literature which has shown various approaches to promote intermediate reasoning steps can improve a final result.

> "role": "user", "content": 'Rate the truthfulness of the following statement: "STATEMENT" Your task is to provide a score from 0 to 100, where

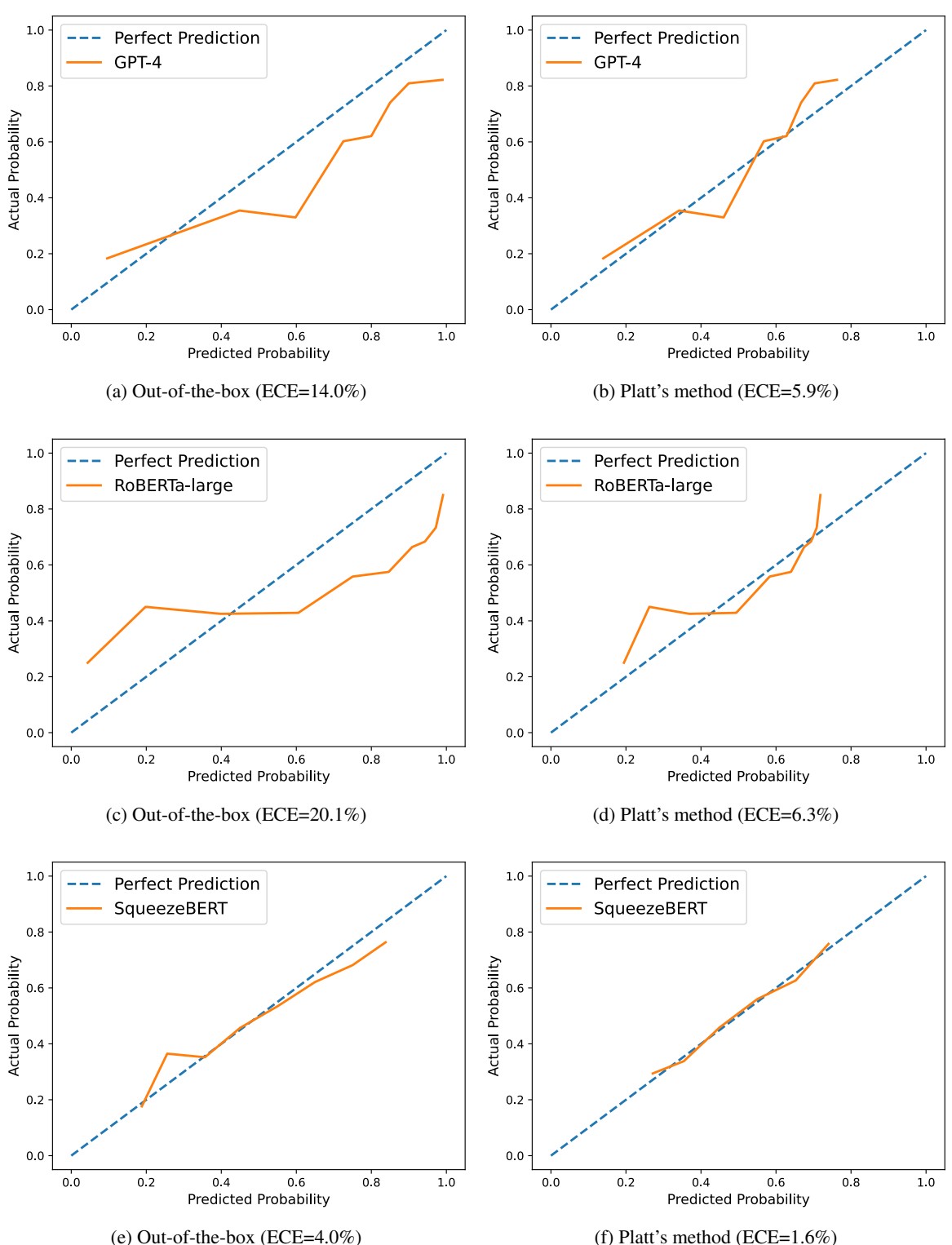

Figure 2: GPT-4 (top), RoBERTa-large (middle), and SqueezeBERT (bottom) probability predictions, out-of-the-box or calibrated with Platt's method.

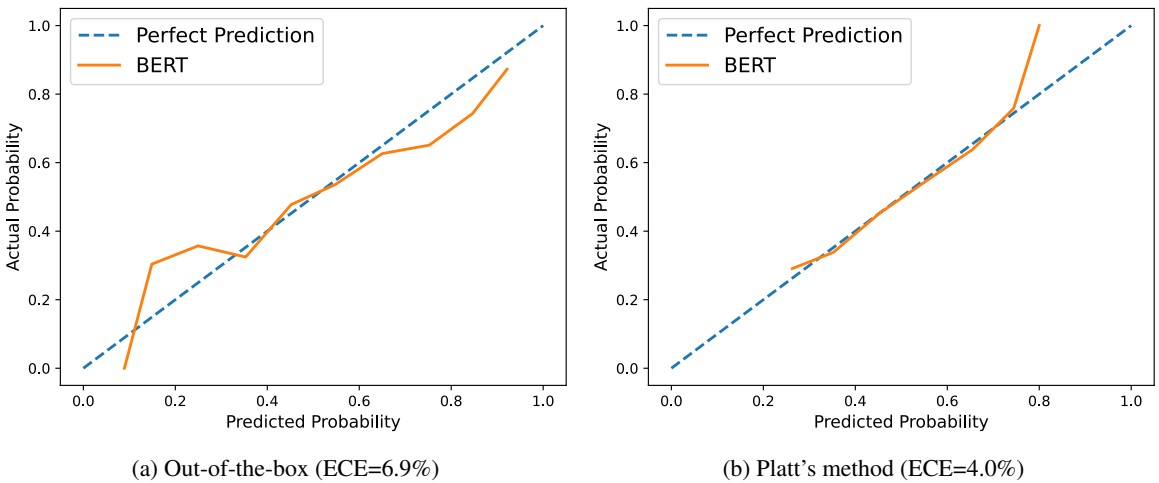

(a) Out-of-the-box (ECE=6.9%)

(b) Platt's method (ECE=4.0%)

Figure 3: BERT probability predictions, out-of-the-box or calibrated with Platt's method.

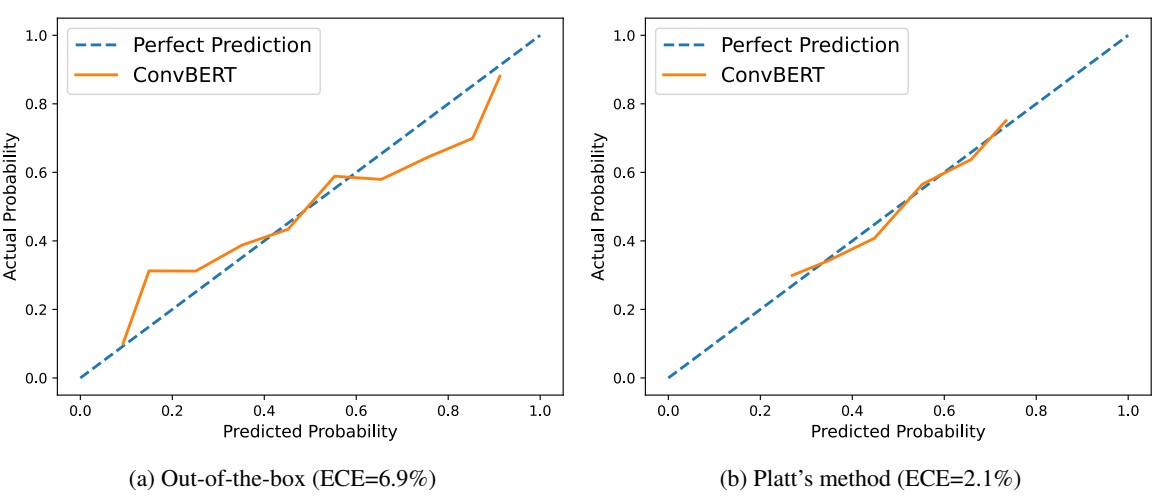

(a) Out-of-the-box (ECE=6.9%)

(b) Platt's method (ECE=2.1%)

Figure 4: ConvBERT probability predictions, out-of-the-box or calibrated with Platt's method.

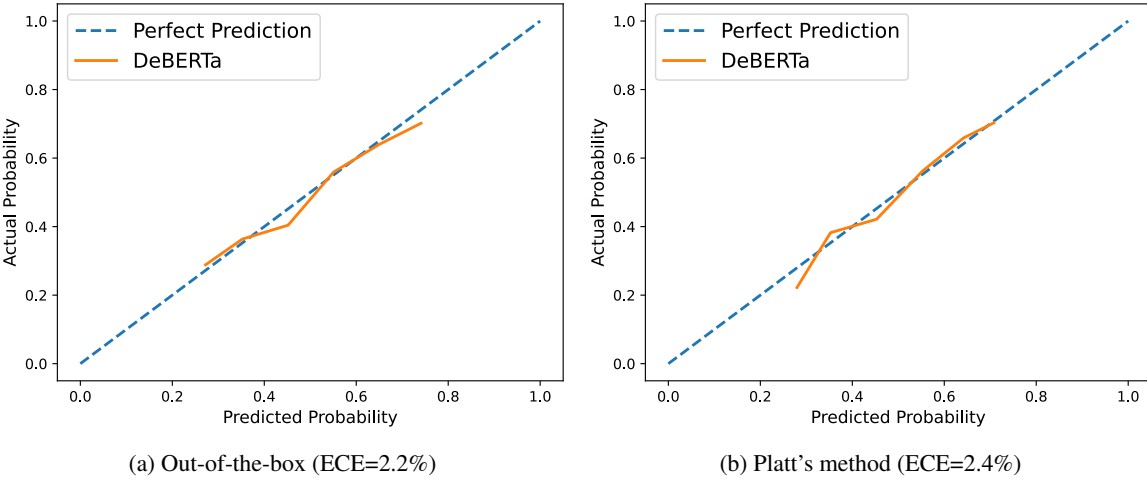

(a) Out-of-the-box (ECE=2.2%)

(b) Platt's method (ECE=2.4%)

Figure 5: DeBERTa probability predictions, out-of-the-box or calibrated with Platt's method.

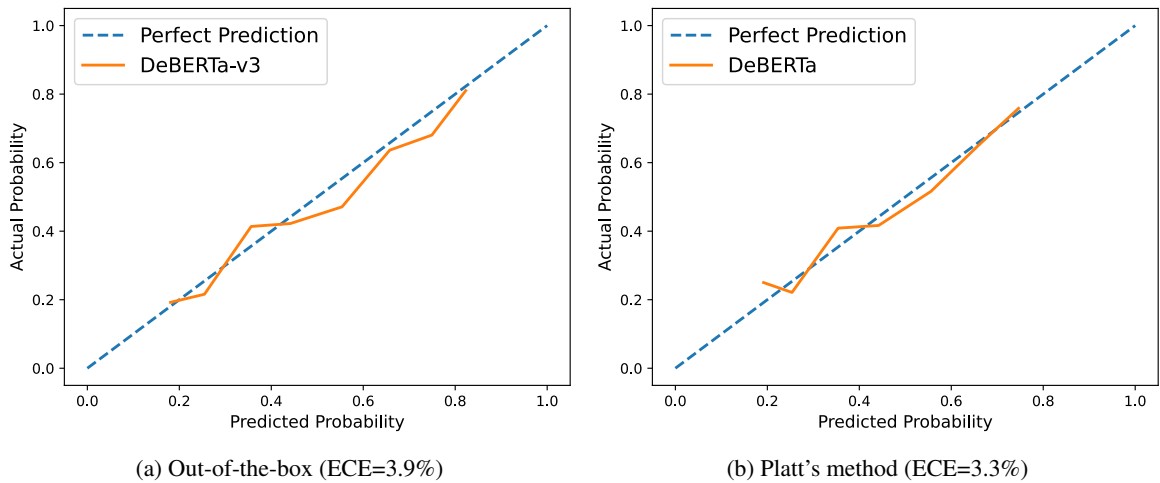

(a) Out-of-the-box (ECE=3.9%)   (b) Platt's method (ECE=3.3%)

Figure 6: DeBERTa-v3 probability predictions, out-of-the-box or calibrated with Platt's method.

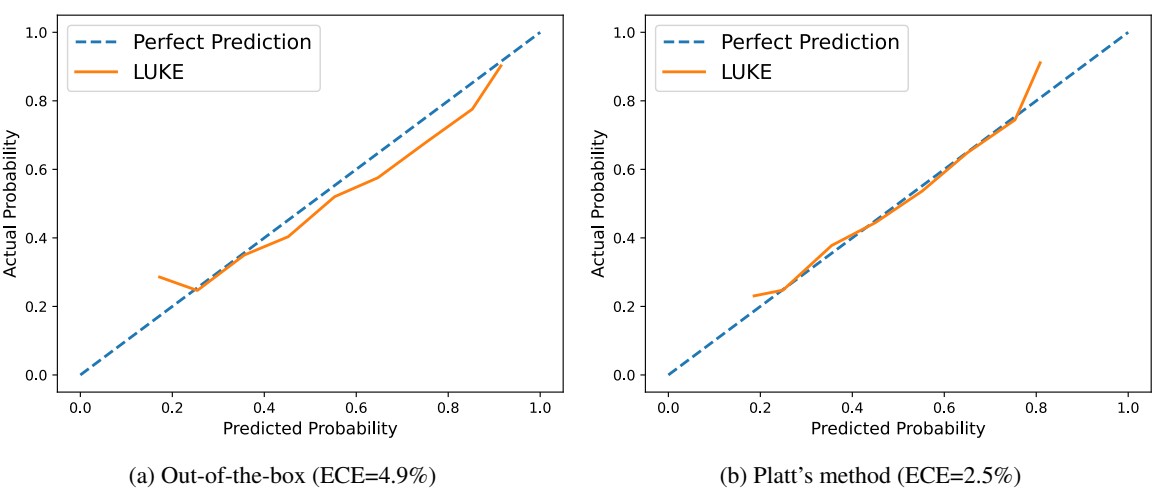

(a) Out-of-the-box (ECE=4.9%)   (b) Platt's method (ECE=2.5%)

Figure 7: LUKE probability predictions, out-of-the-box or calibrated with Platt's method.

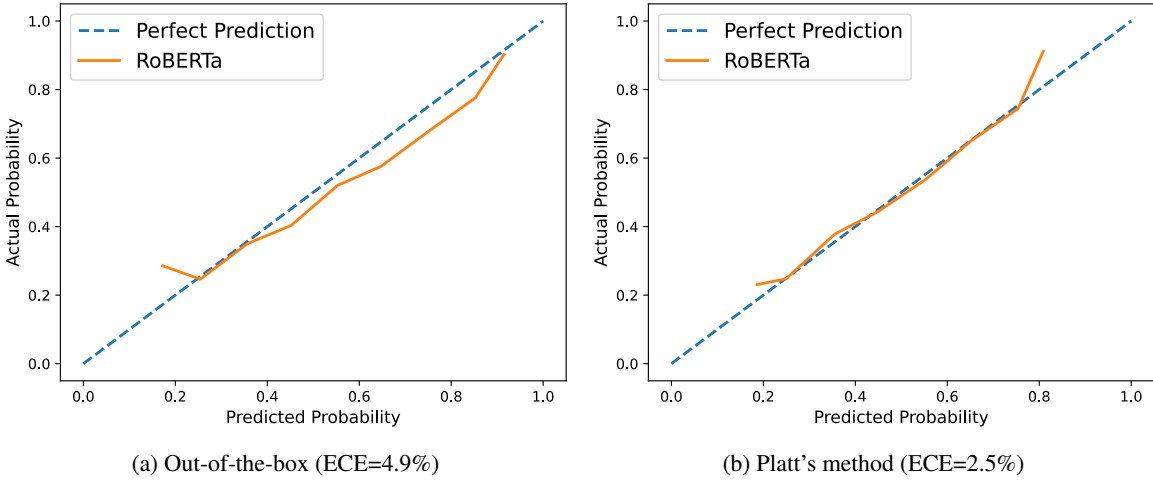

(a) Out-of-the-box (ECE=4.9%)   (b) Platt's method (ECE=2.5%)

Figure 8: RoBERTa probability predictions, out-of-the-box or calibrated with Platt's method.

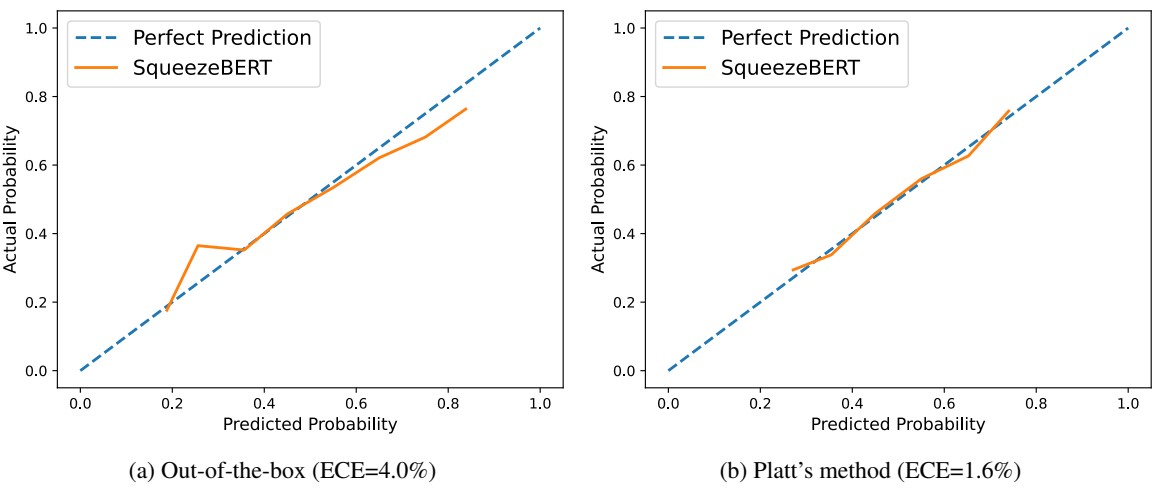

(a) Out-of-the-box (ECE=4.0%)       (b) Platt's method (ECE=1.6%)

Figure 9: SqueezeBERT probability predictions, out-of-the-box or calibrated with Platt's method.

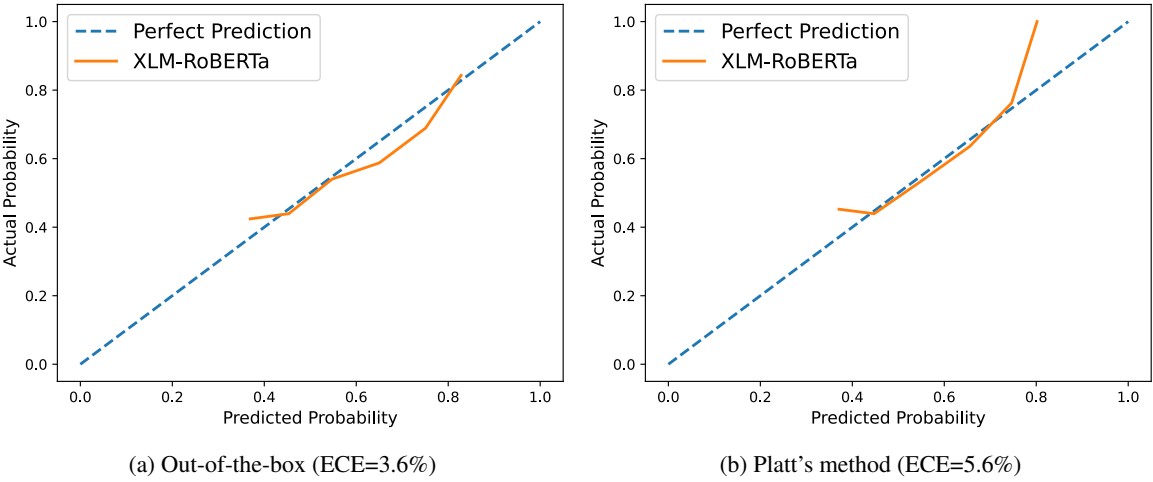

(a) Out-of-the-box (ECE=3.6%)       (b) Platt's method (ECE=5.6%)

Figure 10: XLM-RoBERTa probability predictions, out-of-the-box or calibrated with Platt's method.

0 represents definitively false and 100 represents definitively true, but you must not state your score until you've presented a thorough analysis. Do not begin your response with a number. First write your analysis, then write a vertical bar "|", then finally state your score.'

## Q.2 Predictive Performance

We test the scores these prompts produce compared to no explanation GPT-4 Score Zero-Shot. We report results (the primary metrics for the datasets: accuracy for LIAR, macro F1 for others) in Table 22.

## Q.3 Evaluation of the Explanations: Reasonableness

To perform an initial evaluation of whether the explanations made sense and would be helpful, one of the authors manually examined samples of explanations according to the following directive:

**Definition 4.** *"GT-Reasonable": if the explanation is reasonable in evaluating the statement - in relation to the ground truth PolitiFact or other actual information - please put "y" for "yes". Otherwise, "n" for "no" (e.g., its explanation is false, does not explain the answer, illogical or outright nonsensical, no explanation given at all, etc.).*

We note that this is only a proxy for the ultimate question in explainability: whether the explanation will help a user of an information evaluation system based on this framework. However, that question cannot be answered without knowing the user, a precise definition of "help", etc. This definition ignores all those factors and more, so we do not recommend it for in depth analysis; we use it here simply as a tool to explore the results.

We sampled 30 random examples where GPT-4 gives the correct answer, 10 where it gives the wrong answer, and 10 where it gives the right answer and RoBERTa gives the wrong one. Although we might expect to only get a reasonable explanation if it is giving the right answer, it is conceivable that we could somehow have a correct explanation but a wrong answer. We would also like to better understand why it gives wrong answers. Meanwhile, analyzing the examples GPT-4 got correct and RoBERTa wrong might provide further insights on what GPT-4 relied on to get those answers that RoBERTa could not.

We then did a second round of another 50 examples with the same the same sampling setup, but with the addition of a second evaluation criteria:

**Definition 5.** *"Score-Reasonable": is the explanation reasonable in relation to the score it gives? If so please put "y" for "yes". It would be "n" for "no" if for example it is very confident the statement is true but gave a score around 50 (indicating uncertainty) or well below (indicating false).*

This criteria is quite similar to the previous GT-Reasonable, just making the evaluation compared to the score instead of the ground truth.

We found that 100% of the examples were Score-Reasonable. This is not surprising but confirms the score is stable in relation to the explanation and the model seldom if ever suddenly changes its mind at that stage. We report GT-Reasonable rates in Table 23.

We see that the explanations are generally reasonable when correct. More unexpectedly, they are also often reasonable even when the prediction is wrong. We believe there are two main reasons. First, this often occurs when GPT-4 gives an answer indicating the statement is murky, e.g. score in the 40-60 range, while PolitiFact agrees it is mixed but labels it slightly differently. Due to the threshold at 50, the binary class can end up different, even if both GPT-4 and PolitiFact are largely saying the same thing.

Second, there is some uncertainty and subjectivity in the evaluation. For example, if GPT-4 gives correct framing and general info about the statement, but says it lacks a particular piece of data, is that a good explanation or not? If it gives an explanation with the context of 2021, when really the statement was made in 2008, is that a good explanation?

More refinement of the process is needed. This would likely include more precision in the definition and breaking it down into more categories, as well as larger sample size and more labelers. Nonetheless, these preliminary results already suggest an approach like this can have great potential for explainability, and it is far easier to get here than with previous models like RoBERTa.

We do not recommend training on them, but in case these explanations are useful for further analysis, we publish them on our Github.

## Q.4 Evaluation of the Explanations: Impossible Cases

We took the LIAR-New predictions, filtered for Impossible examples only according to V1 Possibility labels (Appendix C), and randomly sampled

| Dataset | Method | Result |
|---|---|---|
| LIAR | GPT-4 Score-then-Explain | 62.9 |
| LIAR | GPT-4 Explain-then-Score | 64.9 |
| LIAR | GPT-4 Score Zero-Shot | 64.9 |
| LIAR-New All | GPT-4 Explain-then-Score | 65.3 |
| LIAR-New All | GPT-4 Score Zero-Shot | 60.5 |
| CT-FAN English 4-Way | GPT-4 Explain-then-Score | 43.6 |
| CT-FAN English 4-Way | GPT-4 Score Zero-Shot | 42.8 |
| CT-FAN English 3-Way | GPT-4 Explain-then-Score | 63.1 |
| CT-FAN English 3-Way | GPT-4 Score Zero-Shot | 58.6 |

Table 22: Predictive performance of explainability prompts. Explain-then-Score works well, matching or outperforming Score Zero-Shot.

| Sample Type | Round 1 | Round 2 |
|---|---|---|
| Correct | 28/30 | 29/30 |
| Wrong | 5/10 | 8/10 |
| GPT Correct RoBERTa Wrong | 9/10 | 10/10 |

Table 23: GT-Reasonable proportions on examples randomly sampled from each group. Not only are the explanations reasonable in cases it predicted correctly, but also a significant number in which it did not.

50 where GPT-4 made the correct prediction and 50 where it was wrong.

**Correct Answers** Besides cases it does not really know (potentially even saying it does not know) but gets lucky, it seems to either rely on general information or assume a United States context.

The following is an example of relying on general information to evaluate the statement "A Phoenix TV station aired the election results more than a week early."

> The statement refers to a specific event that could be verified through reliable sources. However, without any specific details such as the name of the TV station, the date of the alleged early airing, or the specific election in question, it is difficult to confirm the truthfulness of the claim. Furthermore, it is important to note that the broadcasting of election results before the official end of voting is generally considered unethical and potentially illegal, as it could influence the outcome of the election. Therefore, it would be unusual for a reputable TV station to do so. Finally, it's also worth considering that this could be a misunderstanding

> or misinterpretation of a forecast or prediction being mistaken for actual results.

> Given these considerations, it is impossible to definitively confirm or deny the statement without additional information. | 50

PolitiFact rated this statement "Half True" noting the tv station accidentally aired mock election results, similar to GPT-4's suggestion that it might be a misinterpretation of a "forecast or prediction" (https://www.politifact.com/factchecks/2022/oct/28/instagram-posts/phoenix-tv-station-did-not-call-election-katie-hob/).

The following are examples of assuming a United States context.

**Statement**: "There are 63 million abortions a year in this country."

> The statement asserts that there are 63 million abortions a year in the United States. The Guttmacher Institute, a leading research and policy organization committed to advancing sexual and reproductive health and rights, reported that there were approximately 862,320 abortions in the United States in 2017 [...]

**Statement**: "$2,000 stimulus checks were due on Jan. 19."

> The statement is suggesting that $2,000 stimulus checks were due on Jan. 19. This statement is ambiguous because it does not specify the year or the country it is referring to. However, assuming it refers to the United States and the year 2021, we can analyze its truthfulness [...]

**Statement**: "Our Supreme Court has never taken away a constitutional right."

> The Supreme Court of the United States has the power to interpret the Constitution and determine the constitutionality of laws [...]

Although PolitiFact focuses largely on US politics leading to correct answers here, assuming US context could be a substantial issue if using a system based on this in other countries, or even in the US in the case of evaluating statements that are actually about other countries. While many biases like race, gender, and political ideology are commonly studied in relation to LLMs, this sort of "geographical" bias is less explored, particularly in the misinformation domain. These results suggest it merits further research.

**Incorrect Answers**

In the majority of cases, GPT-4 appears to know it is missing context. In some cases, it seems to assume an image or video which it does not have access to exists, leading to it incorrectly predicting a statement is true.

The following is an example where it knows context is missing when evaluating the statement "A royal guard member collapsed because of the COVID-19 vaccine."

> The statement "A royal guard member collapsed because of the COVID-19 vaccine" is not verifiable without specific information and context. The statement is vague and does not provide enough detail to determine its truthfulness. It doesn't specify which royal guard (from which country or monarchy), nor does it provide any timeline or source of this information. Furthermore, it directly attributes the collapse to the COVID-19 vaccine, which is a complex medical conclusion that would require professional medical evaluation to confirm. Vaccines can have side effects, but serious side effects are rare, and it's also possible for someone to collapse due to other health conditions. Without any specific information or context, it's impossible to definitively confirm or refute this statement. | 50

In a binary evaluation (or even many forms of multiclass evaluation) where a label must be assigned one way or another, errors of this type are not distinguishable from more egregiously wrong and misleading answers. Solutions for such failures may look very different (e.g., querying user or knowledge base for additional information, vs. mitigating hallucinations or overconfidence). So this suggests a need for more uncertainty-aware classification and evaluation methods, such as some of the approaches we propose or hopefully even better future approaches.

The following is an example where GPT-4 seems to take it at face value that images exist, evaluating the statement "Images show how climate change destroyed the Greenland ice sheet and harmed a polar bear from 2009 to 2019."

> The statement is making two claims: 1) climate change has destroyed the Greenland ice sheet and 2) climate change has harmed a polar bear from 2009 to 2019. In conclusion, while the statement captures the essence of the effects of climate change, its use of the term 'destroyed' and the lack of specific evidence regarding the polar bear make it not entirely accurate. | 70

In reality, PolitiFact rates this statement False, explaining "experts pointed out that the image of Greenland has nothing to do with the polar bear image, since polar bears face sea-ice loss in the Arctic Ocean, not Greenland" (https://www.politifact.com/factchecks/2022/jan/19/facebook-posts/10-years-challenge-climate-change-exaggerates-impa/).

Such errors might be addressed by targeted interventions, such as a classifier to detect when an image/video is referenced, making user aware the system cannot see the pictures and might give inaccurate answers, or simply multimodal models that can process pictures smoothly (in theory, GPT-4 might already do this, but this feature is not yet enabled through the API and we were unable to evaluate it).

# R  Web Retrieval

## R.1  Prompt

> "role": "user", "content": 'Rate the truthfulness of the following statement: "STATEMENT" Provide a score from 0 to 100, where 0 represents definitively false and 100 represents definitively true. Do not provide any explanations, only respond with the numerical score.

> The following information may provide additional evidence or context for your rating: ARTICLE'

The first part is the same as our Score prompt, to which we add the last part about "additional evidence or context" and the corresponding PolitiFact article.

### R.2 Answerless Oracle

To remove the final PolitiFact label, we first split the article into sentences. We then iterate backwards through the article looking for a sentence containing one of the keywords "true", "false", or "pants" (for "pants on fire" label). Each of the labels contains one of these, so this locates the last sentence in the article with a label keyword. We then remove that sentence and any subsequent ones. While this may not be a perfect guarantee of removing the label, PolitiFact articles almost always have the verdict at the very end. Generally, the only exception is if the article was later updated, which is not frequent. So this heuristic is quite consistent and sufficient for our preliminary analysis here, which does not aim to say that the performance is precisely X but just shows that sufficiently strong retrieval can massively increase performance compared to not using retrieval.

We make the original PolitiFact articles and this answer-removing code available through our GitHub.