# OpenReview forum: "Towards Reliable Misinformation Mitigation: Generalization, Uncertainty, and GPT-4"
_EMNLP/2023/Conference — EMNLP 2023 Main_

### Official Review · Reviewer_FWg4 · 2023-08-05

**Soundness:** 2

**Excitement:**

3: Ambivalent: It has merits (e.g., it reports state-of-the-art results, the idea is nice), but there are key weaknesses (e.g., it describes incremental work), and it can significantly benefit from another round of revision. However, I won't object to accepting it if my co-reviewers champion it.

**Paper Topic And Main Contributions:**

This paper proposes  a multifaceted investigation to determine if they can use GPT-4, combined with  analysis of generalization and uncertainty, to produce more practical misinformation detection. The paper's aim is to establish a ground  that will lead to practical systems and, improvements in real-world misinformation mitigation. The contributions are as follows:
1-  The study shows that GPT-4 is effective for both hard and soft classification tasks related to misinformation, including multilingual scenarios, outperforming existing approaches.
2 - The research analyzes the generalization capability of GPT-4 compared to previous work.
3 - The study introduces a method to handle uncertainty with GPT-4, which involves excluding examples it struggles to classify effectively.
4 - Introduce a new dataset called LIAR-New consisting of English and French


**Reasons To Accept:**

1) Interesting solution to detect and mitigate misinformation
2) Extensive experiment focused on uncertainty and generalization.
3) Good presentation





**Reasons To Reject:**

1) Technical depth will need to further expanded: The main concern in the paper is that they only discuss this from one point of view of Large Language Models (LLMS) which is GPT-4. The paper claims it is the best which is not true. Google Bard has proven better. Moreover, there are many other LLMS that could be comprised such as LLama or Vicuna. The assumption that GPT-4 is the best out there has significantly weakened the paper.
2) The paper lacks case studies.
3) inefficient survey of existing work.  The paper should be looking for other LLMs and make a valid comparison.


**Reproducibility:**

5: Could easily reproduce the results.

**Reviewer Confidence:**

4: Quite sure. I tried to check the important points carefully. It's unlikely, though conceivable, that I missed something that should affect my ratings.

---

> ### Author Rebuttal · Authors · 2023-08-29
>
> > Reviewer: Technical depth will need to further expanded: The main concern in the paper is that they only discuss this from one point of view of Large Language Models (LLMS) which is GPT-4. The paper claims it is the best which is not true. Google Bard has proven better. Moreover, there are many other LLMS that could be comprised such as LLama or Vicuna. The assumption that GPT-4 is the best out there has significantly weakened the paper.
>
>
> We agree that GPT-4 is not universally strongest, and it is valuable to test other LLMs. We will amend the paper to only state that GPT-4 is the strongest in many tasks. Furthermore, we have now tested some other models. Specifically:
>
> LIAR:
> Llama 2 13b (instruct-tuned): 52.4 (accuracy)
> Llama 2 70b (instruct-tuned): 50.1 (accuracy)
> PaLM 2 Bison: 62.4 (accuracy)
> Old result: GPT-4 Score Zero-Shot: 64.9 (accuracy)
>
> CT-FAN-22 English:
> Llama 2 13b (instruct-tuned): 22.3 (macro f1)
> Llama 2 70b (instruct-tuned): 21.4 (macro f1)
> PaLM 2 Bison: 21.3 (macro f1)
> Old result: GPT-4 Score Zero-Shot: 42.8 (macro f1)
>
> We also tested Bison on CT-FAN German: 22.3 (macro f1). These results correspond to our Table 2 and are the “Score Zero-Shot” accuracies by the terminology of the paper. Unfortunately, tuning the score threshold does not enable any significant improvement, for example, on LIAR Bison’s performance is already maximized, and at best Llama’s performance increases to 56.4 by predicting majority class only. Unfortunately, Bard itself is not available in our country for testing, and we similarly were not able to access PaLM 2 Unicorn which powers Bard (Bison reported here is the second largest version, while Unicorn is the largest).
>
> Although Bison does not perform as well as GPT-4, and may particularly struggle with the longer and more complicated inputs of CT-FAN-22, on LIAR it is close to the other language models tested, without relying on a training set and without being the strongest PaLM 2 model. So, this again highlights the potential of recent LLMs for this domain. We will add these results to our paper.
>
> We chose to start with GPT-4 because Bard was not accessible in our country, and most resources we have seen indicate GPT-4 is strongest. The comparisons between PaLM 2 (on which Bard is based) and GPT-4 highlighted in Table 5 of the PaLM 2 Technical Report, where PaLM 2 wins 2 out of 3, omitted a number of other comparisons. For example, GPT-4 has superior performance on MMLU https://paperswithcode.com/sota/multi-task-language-understanding-on-mmlu, but the PaLM 2 report only gave their own result there with no comparison. Similarly, GSM8K  https://paperswithcode.com/sota/arithmetic-reasoning-on-gsm8k, HellaSwag https://paperswithcode.com/sota/sentence-completion-on-hellaswag, and HumanEval https://paperswithcode.com/sota/code-generation-on-humaneval.
>
> We would also like to highlight that should Bard or another model prove better than GPT-4 here in the future, that does not detract but actually strengthens the value of our results. That is, if recent LLMs can do even better than the results we have found so far, that means they will be providing even more value in terms of performance, practical quantification of uncertainty, and generalization.
>
>
> > Reviewer: The paper lacks case studies.
>
> A case study was presented in Appendix F (“LIAR Error Analysis Case Study”). We look at examples GPT-4 gets wrong but RoBERTa gets right, and show how some examples are extremely close semantically to training set examples. In these cases, learning from the training set examples could be helpful, but could also result in artificially easy test set examples that don’t reflect generalization.
> We will also add case studies on explainability as discussed in the response to reviewer nBWy. If those do not cover the type of case study you have in mind, we would welcome additional feedback here.
>
> > Reviewer: inefficient survey of existing work. The paper should be looking for other LLMs and make a valid comparison.
>
> The paper compares extensively with the state-of-the-art on these tasks and datasets. This state-of-the-art was not other recent (2023, generative) LLMs, it was older models such as  BERT and RoBERTa. Certainly other LLMs could be explored and we make no argument that other recent LLMs could not give better results than GPT-4 in this domain. The paper also suggests other recent LLMs as a direction for future research, which we have been looking into after submitting the paper and have shared some of the results in response to the previous question.
>
> We will also add a section to the literature review on other recent LLMs, and discuss which ones like Bard would be most promising for future investigation. If you have references showing that Bard is the strongest overall model available (besides the PaLM 2 technical report, which as we discussed above seems insufficient), we would be very happy to learn from them and include them. More generally, if you have any other insights on which models might have the most potential in this domain and why (and therefore would provide the most value as suggestions for future researchers to investigate), those would likewise be greatly appreciated.

---

### Official Review · Reviewer_H7JA · 2023-08-05

**Typos Grammar Style And Presentation Improvements:** 1. Section 3.1 is a bit too long. It'…
**Soundness:** 5

**Excitement:**

5: Transformative: This paper is likely to change its subfield or computational linguistics broadly. It should be considered for a best paper award. This paper changes the current understanding of some phenomenon, shows a widely held practice to be erroneous in someway, enables a promising direction of research for a (broad or narrow) topic, or creates an exciting new technique.

**Paper Topic And Main Contributions:**

Instead of technical contributions,
this paper studies the feasibility of using GPT-4 on misinformation prediction.
This paper conducts extensive experiments on three different datasets in English and French.
It analyzes the generalization and uncertainty of GPT-4.
The results show GPT-4 can be a better misinformation predictor than prior work. It also finds that GPT-4 has a different failure mode from other transformer-based LMs.

**Reasons To Accept:**

1. This paper is well-organized and easy to read.
2. The experiments are quite extensive with different settings and baseline methods. The experimental result analysis is inspiring. These can well support the research claims and conclusions.
3. This paper proposes a new multilingual dataset on misinformation prediction.

**Reasons To Reject:**

1. Prompts exploration is insufficient.

    Although this paper proposes different prompts for misinformation prediction,
    it does not try other prompts to enable in-context learning with samples from a training set or human-crafted.
    It is expected to see how this can further improve the ability of GPT-4 on misinformation prediction.

**Reproducibility:**

3: Could reproduce the results with some difficulty. The settings of parameters are underspecified or subjectively determined; the training/evaluation data are not widely available.

**Reviewer Confidence:**

5: Positive that my evaluation is correct. I read the paper very carefully and I am very familiar with related work.

---

> ### Author Rebuttal · Authors · 2023-08-29
>
> ## Techniques to improve performance
>
> We agreed there could potential for improvement - perhaps significant improvement - with better prompting strategies. We’ve now run a number of experiments with different prompts, along with other techniques to improve the performance. This includes two explainability prompts, three prompts for in-context learning with automatically chosen demonstration, finetuning GPT-3.5 on the LIAR training set (and 8 other new results with GPT-3.5), and using PolitiFact articles to evaluate the potential of web retrieval systems in this context.
>
> Most of these approaches provided only marginal improvements, or directly made performance worse. Only web retrieval yielded better results, and shows potential for very large performance improvements - 20+ percentage points, with a sufficiently strong retrieval system. Based on these results, we believe that is the most promising direction for reducing errors (in tandem with other directions like explainability and improved uncertainty quantification, that are critical for real-world results but not directly aimed at reducing errors).
>
> Below we discuss all of these results in more detail.
>
>
>
> ### Classification Performance of Explainability Prompts:
>
> First, we consider two prompts related to explainability, also discussed in response to Reviewer nBWy:
>
> Prompt 1, (Score Zero-shot): 64.9 (accuracy)
> > Rate the truthfulness of the following statement: "{text}" \
> > Your task is to provide a score from 0 to 100, where 0 represents definitively false and 100 represents definitively true, but you must not state your score until you've presented a thorough analysis. Do not begin your response with a number. First write your analysis, then write a vertical bar "|", then finally state your score.
>
> Prompt 2, (Score Zero-shot): 62.9 (accuracy)
> > Rate the truthfulness of the following statement: "{text}" \
> > Provide a score from 0 to 100, where 0 represents definitively false and 100 represents definitively true. Then write a vertical bar "|", followed by explaining your reasoning for the score.
>
> The first prompt, which has some chain-of-thought characteristics in terms of asking for reasoning first, gives results equal to the Score prompt in the paper, while the second’s performance is a bit worse.
>
>
> ### In-context learning with an automatically chosen demonstration:
>
> The following three prompts use a demonstration chosen by embedding similarity of the target statement with statements in the training set. This has similarities to https://arxiv.org/abs/2305.14128 and is particularly motivated by the observations in section 4.5 of our paper, i.e. that GPT-4 and RoBERTa make different errors in a way that is tied to embedding similarity with training set examples. The hope is that by providing the most similar training set example, that provides both a demonstration and potentially a direct source of information relevant to the target example. Unfortunately, however, this proved ineffective.
>
> In the below prompts, closest_train_label refers to the label of the closest training example, uniformly mapped onto the 0-100 range. Closest_train_text refers to the corresponding input statement.
>
> Prompt 3, (Score Zero-shot): 57.2 (accuracy)
> > Rate the truthfulness of the following statement: "{statement}"
>
> > Here is the rating of a similar statement. The rating is accurate, but the statement may or may not be relevant to this one.
> > Similar statement: "{closest_train_text}"
> > Rating: {closest_train_label}
>
> > Provide a score for the original statement from 0 to 100, where 0 represents definitively false and 100 represents definitively true. Do not provide any explanations, only respond with the numerical score.
>
> Prompt 4, (Score Zero-shot): 64.2 (accuracy)
> > Here is a similar statement that is {closest_train_label}: "{closest_train_text}”
> > Now rate the truthfulness of the following statement: "{statement}”
> > Provide a score from 0 to 100, where 0 represents definitively false and 100 represents definitively true. Do not provide any explanations, only respond with the numerical score.'
>
>
> Prompt 5, (Score Zero-shot): 65.2 (accuracy)
> > If the most similar training set example is in the top 10% most similar: Prompt 4
> > Else: original Score prompt
>
> Prompt 3 doesn’t work well, but 4 and 5 are OK. 5 gives an improvement on GPT-4 Score Zero-Shot (64.9), but 0.3% is marginal - could easily be noise, and not likely to be worth the extra tokens and complications.
>
>
> ### Finetuning and GPT-3.5
>
> While not exactly prompting, in the similar vein of general strategies to improve a foundation model’s performance, we also tested GPT-3.5 and its newly-released finetuning feature. Specifically, we use the following prompt:
>
> >Rate the truthfulness of the following statement: "{statement}" \
> >Provide a score from 0 to 100, where 0 represents definitively false and 100 represents definitively true.
>
> This is the same as the usual Score prompt, except the removal of the last sentence “Do not provide any explanations, only respond with the numerical score.” We remove this because the finetuning should already be sufficient showing it to respond only with the numerical score (and in practice it does, 100% of the time). We map the original 6-way LIAR labels to scores 0-100 uniformly, i.e. [0,20,40,60,80,100]. Then we finetune with default settings, which resulted in 2 epochs of finetuning. We compare with the untuned version with the usual Score prompt. The results on LIAR were:
>
> GPT-3.5 Score Zero-Shot: 67.3 (accuracy)
> GPT-3.5 Score Zero-Shot Finetuned: 57.2 (accuracy)
>
> The finetuning is not effective and seems to damage the generalization ability. We hypothesize that finetuning to Impossible examples might be particularly damaging. That said, this is a first attempt using a system that was only released last week, so there is considerable room for experimentation that might yield better results. Further research in this direction is needed, as well as testing GPT-4 finetuning when that becomes available (later this year according to OpenAI).
> We note that “untuned” GPT 3.5 performs surprisingly well, better than GPT 4 in the same Score Zero-Shot setup. However, there are some caveats. First, its score is already maximized at the zero-shot binarization threshold of 50, meaning it cannot achieve higher performance through tuning on the validation set like GPT-4 can. The margins are small, however, so this may not have a big impact. More significantly, the performance seems brittle, as illustrated by the following results:
>
> LIAR GPT-3.5 Binary: 53.6 (accuracy)
> LIAR-New “All” GPT-3.5 Binary: 55.7 (macro f1)
> LIAR-New “Possible” GPT-3.5 Binary: 58.3 (macro f1)
> LIAR-New “All” GPT-3.5 Score Zero-Shot: 61.1 (macro f1)
> LIAR-New “Possible” GPT-3.5 Score Zero-Shot: 60.3 (macro f1)
> LIAR-New “All” GPT-3.5 Binary Uncertainty-Enabled: 67.7 (macro f1)
> LIAR-New “Possible” GPT-3.5 Binary Uncertainty-Enabled: 67.5 (macro f1)
>
> In addition, in the Uncertainty-Enabled setting, it marked 794 examples as ones it did not have certainty on, not too far from GPT-4’s 914. Of which 519 (65.4%) were Impossible, which is a bit lower than GPT-4’s 69.1% but again comparable. And we do see a relative performance increase. But in terms of the absolute numbers, in all 7 of these cases, the performance is much worse compared to GPT-4. Therefore, GPT-3.5’s strong performance on LIAR with Score Zero-Shot prompt may be mostly luck.
>
>
> ### Web Retrieval
>
> Another route to improving performance is incorporating external information. We conducted an initial investigation of this approach. In particular, as part of the collection process of LIAR-New, in addition to the statements and labels, we also obtained all of the corresponding PolitiFact articles. We set up the following prompt:
>
> >Rate the truthfulness of the following statement: "{input_text}"
> >Provide a score from 0 to 100, where 0 represents definitively false and 100 represents definitively true. Do not provide any explanations, only respond with the numerical score.
>
> The following information may provide additional evidence or context for your rating:
> {article_text}
>
> We consider two settings. First, we just provide the entire PolitiFact article. We call this “Web Oracle” because it represents nearly the ideal case for web retrieval - the document is perfectly relevant and directly gives veracity judgment. Second, we remove the last part of the PolitiFact article where PolitiFact’s rating is stated. Here the document is still perfectly relevant, but the system has to do more evaluation of the information presented itself.
>
> LIAR-New:
> GPT-4 Web Oracle: 98.6 (macro f1)
> GPT-4 Web Answerless Oracle: 90.7 (macro f1)
>
> This shows an enormous performance improvement compared to all other approaches. Now, there is a significant limitation: here we already had and knew where to get relevant information. This is a standard approach for developing and evaluating parts of an overall system involving information retrieval (e.g., it is used in the well-known FEVER dataset paper https://aclanthology.org/N18-1074/), but not possible in practice. So more research is needed here to reach these results with non-oracle retrieval systems. Nonetheless, these results show there is enormous potential for these systems to use external information from such systems.
>
>
> ### Overall
>
> We hope all of these results will provide some further insights on prompting and other strategies to improve performance of a fixed foundation model in this context. We recognize that there are many other techniques still unexplored, such as more classic chain-of-thought, few-shot setups with more than one example or with the examples chosen in different ways, and many more. However, after considering the results here where most changes had either limited or negative impact, we believe it may still be too soon to devote too much research to fully optimizing prompting for performance here. Rather, we suggest that first research should focus on integrating modules necessary for strong real-world results from such a system, such as web retrieval which provides a route to much larger performance increases than prompting, and explainability to make predictions that inspire greater confidence and provide more practical information. Then when a system is close to ready for real-world use, or when there are no more promising modules to add, that will be the most efficient time to thoroughly explore prompts. Otherwise, there is a substantial risk that carefully crafted prompts will be rendered obsolete by new modules that greatly change how the system works.
>
> We will add these experiments, results, and discussion to the paper.
>
>
>
>
> ## Reviewer: Section 3.1 is a bit too long. It's better to add some subsections.
>
> Thank you for the feedback, we will add subsections separating the prompts and the other information on converting the GPT-4 answers to label predictions.
>
> ## Reviewer: The exact expenses on OpenAI APIs could be included as well.
>
> For GPT-4 Score Zero-Shot on LIAR, the cost is approximately 3.2 USD (100k prompt tokens, 3k completion tokens). On LIAR-New, 4.6 USD (154k, 2k). On CT-FAN English, where the input examples are much longer,  21.8 USD (725k, 600). On CT-FAN German, 28.7 USD (956k, 700). These numbers can vary slightly between runs of the model (depending on completion tokens), as well as slight differences in the three prompts in the paper, but should provide reasonable estimates.
>
> Among the new experiments, Prompt 1 above (adding explanations to the output instead of just the number) cost 23.1 USD (142k, 314k). The finetuning on the LIAR train set comprised 609k tokens costing 4.9 USD. Web Retrieval Oracle cost 53 USD (1765k, 2k).
>
> We see that there is a very large range depending on different data and prompts. Furthermore, the overall budget needed for these experiments is not trivial. In the short-term, thoughtful prompting and choice of experiments to target can help keep costs down when budget is limited. In the longterm, cheaper models with comparable performance will almost certainly be found or created, which will be a significant advantage for both research and real-world deployment.
>
> We will add these numbers and discussion to the paper.

---

### Official Review · Reviewer_nBWy · 2023-08-06

**Soundness:** 4

**Excitement:**

4: Strong: This paper deepens the understanding of some phenomenon or lowers the barriers to an existing research direction.

**Paper Topic And Main Contributions:**

This work proposes a new dataset for fake news detection in English and French. It demonstrate that GPT-4 outperform existing methods in fake news detection, revealing that GPT-4 and RoBERTa-large exhibit critical differences in failure models. Moreover, the code and dataset are publicly available for the community to reproduce the results and conduct further research.

**Reasons To Accept:**

1. Dataset Creation and Uncertainty Handling: The proposed LIAR-New dataset with paired English and French misinformation data, along with Possibility labels indicating context for veracity evaluation, represents a valuable resource for the research community. Additionally, the proposed techniques to handle uncertainty demonstrate the paper's practicality and applicability in real-world settings, where perfect classification is often challenging.

2. Comparative Performance and Novel Findings: GPT-4, as the most advanced LLM by OpenAI, have demonstrated excellent performance on many real-life tasks. The results in this work demonstrates that GPT-4 can outperform existing methods in various settings and languages, showcasing the superiority of the proposed approach. The exploration of generalization differences between GPT-4 and RoBERTa-large offers novel insights that hold promise for significant performance improvements.

These dataset creation efforts, the uncertainty handling techniques and the results with GPT-4 are important contributions that will drive progress in misinformation detection and mitigation.

**Reasons To Reject:**

1. In the GPT-4 experiments, the instructions provided explicit prompts to the model, specifying the omission of explanations in the returned outputs, and only scores were requested. However, it is worth noting that explanations could potentially offer valuable insights for interpreting the results to some extent. The rationale behind discouraging the inclusion of explanations in the outputs is not mentioned in the paper.

**Reproducibility:**

5: Could easily reproduce the results.

**Reviewer Confidence:**

3: Pretty sure, but there's a chance I missed something. Although I have a good feel for this area in general, I did not carefully check the paper's details, e.g., the math, experimental design, or novelty.

---

> ### Author Rebuttal · Authors · 2023-08-29
>
> The main rationale for explicitly discouraging explanations was to make sure the results could be evaluated quantitatively in a direct way. Without “Do not provide any explanations, only respond with the number“ in the prompt, the responses will not follow a set format, making it difficult to reliably extract the prediction, especially if it includes multiple numbers in the response. Potentially a more complicated prompt that asks for a specific formatting could work, but our approach also reduces cost and increases speed by minimizing output tokens. The cost difference is also quite substantial, approximately 3.2 vs. 23.1 USD per experiment on LIAR (which is token-wise the smallest and therefore cheapest of the datasets - please see also bottom of response to reviewer H7JA). We will add a note clarifying this in the paper.
>
> That said, we agree that explanations are valuable. The paper notes this as a promising direction for future work, but we are excited to share some preliminary results. We changed our prompt to the following:
>
> > Rate the truthfulness of the following statement: "{text}" \
> > Your task is to provide a score from 0 to 100, where 0 represents definitively false and 100 represents definitively true, but you must not state your score until you've presented a thorough analysis. Do not begin your response with a number. First write your analysis, then write a vertical bar "|", then finally state your score.
>
> This prompt is inspired by https://arxiv.org/abs/2308.00071 which suggested asking for explanation first followed by final answer is more effective than the reverse, as well as more general chain-of-thought literature.
>
> We applied the prompt to LIAR, then using the original PolitiFact articles that LIAR is derived from, we evaluated the reasonableness of the explanations. Specifically, one of the authors went through each example manually and labeled them according to the following:
>
> > "GT-Reasonable": if the explanation is reasonable in evaluating the statement - in relation to the ground truth PolitiFact or other actual information - please put "y" for "yes". Otherwise, "n" for "no" (e.g., its explanation is false, does not explain the answer, illogical or outright nonsensical, no explanation given at all, etc.).
>
> We note that this definition is only a proxy for the ultimate question in explainability here: whether the explanation will help a user of an information evaluation system based on this framework. However, that question cannot be answered without knowing the user, a precise definition of “help”, etc.
>
> We sampled 30 random examples where GPT-4 gives the correct answer, 10 where it gives the wrong answer, and 10 where it gives the right answer and RoBERTa gives the wrong one. This division is motivated that mainly we only expect a reasonable explanation if it is giving the right answer, but we also might have a correct explanation but (somehow) a wrong answer, or in general want to understand what is happening in the wrong answers for better interpretation of the results like you highlighted. Meanwhile, analyzing the ones GPT-4 got correct and RoBERTa wrong might provide further insights on what GPT-4 relied on to get those answers that RoBERTa could not.
>
> We then did a second round of another 50 examples with the same the same sampling setup, but with the addition of a second evaluation criteria:
>
> > “Score-Reasonable”: is the explanation reasonable in relation to the score it gives? If so please put “y” for “yes”. It would be "n" for "no" if for example it is very confident the statement is true but gave a score around 50 (indicating uncertainty) or well below (indicating false).
>
> This criteria is quite similar to the previous GT-Reasonable, just evaluating compared to the score instead of the ground truth.
>
> ## Results:
>
> Round 1 GT-Reasonable rates:
> Random correct: 28/30
> Random wrong: 5/10
> Random GPT correct RoBERTa wrong: 9/10
>
> Round 2 GT-Reasonable rates:
> Random correct: 29/30
> Random wrong: 8/10
> Random GPT correct RoBERTa wrong: 10/10
>
> Round 2 Score-Reasonable rates:
> Random correct: 30/30
> Random wrong: 10/10
> Random GPT correct RoBERTa wrong: 10/10
>
> We see that the explanations seem quite reasonable. There are no issues with score matching to explanation (Score-Reasonable rates = 100%), and even compared to the ground truth most explanations appear to make sense. Most surprisingly, this holds even for the case where GPT-4’s prediction is wrong. We two main reasons:
> This often happens when GPT-4 gives an answer indicating the statement is murky, e.g. score in the 40-60, which PolitiFact also says but labels slightly differently. Due to the threshold at 50, the binary class can end up different, even if both GPT-4 and PolitiFact are largely saying the same thing.
> There is some uncertainty in the evaluation. For example, if the LLM gives correct framing and general info about the statement, but says it lacks a particular piece of data, is that a good explanation or not? If it gives an explanation with the context of 2021, when really the statement was made in 2008, is that a good explanation?
>
> More refinement of the process, likely breaking it down into more categories (as well as more samples and more labelers), is needed. Nonetheless, these preliminary results already suggest an approach like this can have great potential for explainability, and it is far easier to get here than with previous models like RoBERTa. We will add this experiment to the paper, along with publishing the data and adding some case studies to the appendix. We hope it will provide a launchpad for in depth future work on explainability in this context.

---

### Meta-Review · Area_Chair_M94K · 2023-09-18

**Recommendation:** 5

**Metareview:**

This paper provides a new dataset for misinformation detection and mitigation in French and English. The authors show that GPT-4 outperforms existing methods and that GPT-4 and RoBERTa show large differences in what they fail to detect. They further show that training GPT-4 with an uncertainty-based method that excludes examples that are difficult to classify further improves performance. The dataset and code will be publicly released and should be useful for the community. Some of the weaknesses regarding the explanations and prompts were sufficiently addressed by the authors in the rebuttal and the results and analysis would be great to add to the paper. One reviewer had concerns about the number of different LLMs tested. The authors responded with some insight into other models and I believe that this is appropriate for the scope of this work. Overall, the work appears to be sound and exciting.

---

### Decision · Program_Chairs · 2023-10-07

**Decision:**

Accept-Main

**Comment:**

This paper provides a new dataset for misinformation detection and mitigation in French and English. The authors show that GPT-4 outperforms existing methods and that GPT-4 and RoBERTa show large differences in what they fail to detect. They further show that training GPT-4 with an uncertainty-based method that excludes examples that are difficult to classify further improves performance. The dataset and code will be publicly released and should be useful for the community. Some of the weaknesses regarding the explanations and prompts were sufficiently addressed by the authors in the rebuttal and the results and analysis would be great to add to the paper. One reviewer had concerns about the number of different LLMs tested. The authors responded with some insight into other models and I believe that this is appropriate for the scope of this work. Overall, the work appears to be sound and exciting.